# Causal de Finetti: On the Identification of Invariant Causal Structure in Exchangeable Data

**Siyuan Guo**[12*]    **Viktor Tóth**[1*]    **Bernhard Schölkopf**[2]    **Ferenc Huszár**[1]

[1]University of Cambridge    [2]Max Planck Institute for Intelligent Systems

{syg26,fh277}@cam.ac.uk    toth.viktor7400@gmail.com
bs@tuebingen.mpg.de

## Abstract

Constraint-based causal discovery methods leverage conditional independence tests to infer causal relationships in a wide variety of applications. Just as the majority of machine learning methods, existing work focuses on studying *independent and identically distributed* data. However, it is known that even with infinite i.i.d. data, constraint-based methods can only identify causal structures up to broad Markov equivalence classes, posing a fundamental limitation for causal discovery. In this work, we observe that exchangeable data contains richer conditional independence structure than i.i.d. data, and show how the richer structure can be leveraged for causal discovery. We first present causal de Finetti theorems, which state that exchangeable distributions with certain non-trivial conditional independences can always be represented as *independent causal mechanism (ICM)* generative processes. We then present our main identifiability theorem, which shows that given data from an ICM generative process, its unique causal structure can be identified through performing conditional independence tests. We finally develop a causal discovery algorithm and demonstrate its applicability to inferring causal relationships from multi-environment data. Our code and models are publicly available at: https://github.com/syguo96/Causal-de-Finetti

## 1   Introduction

Learning causal structure from observational data is a key step towards causally robust predictions in machine learning. Most existing causal discovery theory (Pearl, 2009) focuses on the study of *independent and identically distributed (i. i. d.)* data. Indeed, a majority of practical methods (Chickering, 2002; Spirtes et al., 2000a,b) based on i.i.d. data only allows us to determine causal structure up to broad equivalence classes, and going beyond that is known to be impossible without further constraints (Pearl, 2009). For example, the basic task of identifying a bivariate cause-effect relationship (i.e. $X$ causes $Y$ or $Y$ causes $X$) on i.i.d. data is known to be impossible. Current methods impose additional restrictions, e.g., linearity assumptions (Hoyer et al., 2012; Shimizu et al., 2006, 2011) or assumptions about the additive nature of noise (Hoyer et al., 2008a; Peters et al., 2014; Zhang and Hyvärinen, 2009) to ensure identification.

A more recent line of work relaxes the i.i.d. assumption and considers inferring causal structure from grouped or multi-environment data (Arjovsky et al., 2019; Heinze-Deml et al., 2018; Huang et al., 2020; Peters and Meinshausen, 2016; Rojas-Carulla et al., 2018; Tian and Pearl, 2001). Our key observation is that studying grouped data is akin to relaxing the assumption on the data generating process from i.i.d. to exchangeable. In this work, we study causal structure learning in exchangeable data and show unique causal structure identification is enabled by the richer conditional independence structure of exchangeable processes.

---

[*]Equal contribution

37th Conference on Neural Information Processing Systems (NeurIPS 2023).

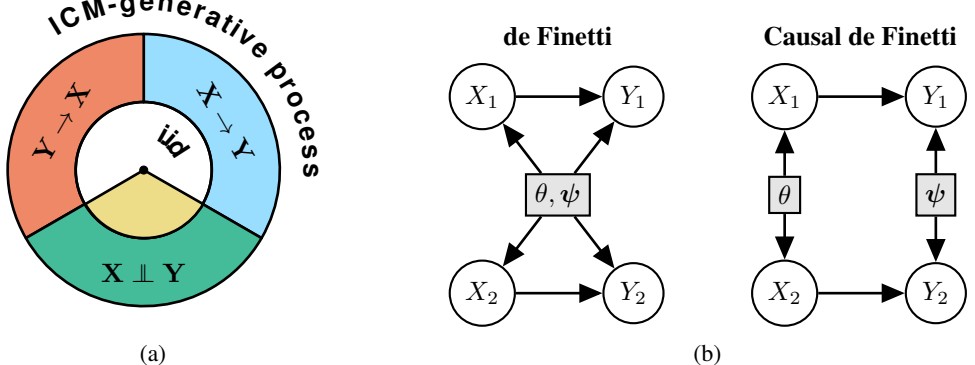

Figure 1: (a) is an illustration showing how i.i.d. data and certain exchangeable data differ in identifying the correct causal structure for a bivariate model. Each quadrant represents a causal structure, $X \perp\!\!\!\perp Y, X \rightarrow Y, X \leftarrow Y$. The inner circle represents i.i.d. regime and the outer circle represents certain exchangeable regime. Under i.i.d. data, one can only identify $X \perp\!\!\!\perp Y$, whereas certain exchangeable data (i.e., ICM-generative processes) enables one to identify unique causal structures. (b) illustrates a differentiating factor between de Finetti and causal de Finetti's representation on exchangeable data. Causal de Finetti disentangles the latents and substantiates causal mechanisms are independent in the sense latent parameters governing each mechanisms are statistically independent.

Several works in multi-environment data implicitly leverage the *independent causal mechanism (ICM)* assumption (Aldrich, 1989; Janzing and Schölkopf, 2010; Pearl, 2009), which postulates that causal mechanisms of the true underlying generating process do not inform or influence one another. Despite having been widely applied (Brehmer et al., 2022; Goyal et al., 2019; Madan et al., 2021; Parascandolo et al., 2018), the ICM assumption is rarely stated any more formally than above, and thus lacks a statistical formalization. It is also unknown what the fundamental limitations of inferring causal structure under the ICM assumptions are. Our work makes three contributions to clarify these questions:

- **Causal de Finetti theorems** (§ 3) provide a mathematical justification for the *independent causal mechanism (ICM)* assumption in data generating processes. It states that any exchangeable process satisfying a certain set of conditional independence properties can be represented as a generative process in which factors in a Markov factorization are statistically independent. We show how causal de Finetti substantiates the ICM principle and call such models ICM-generative processes.

- **Our main identifiability theorem** (§ 4), informally stated, shows that if data is sampled from an ICM generative process, the causal graph is uniquely identifiable by testing conditional independence relationships.

- **Causal discovery in multi-environment data:** Section 5 connects the identifiability theorem for ICM generative models to the analysis of multi-environment data. This section establishes that multi-environment data can be viewed as observing finite marginals of i.i.d. copies of an exchangeable process. This then allows us to develop an algorithm for recovering causal structure from data coming from a sufficient number of environments.

Our work thus provides strong probabilistic justification for approaches based on the independent causal mechanisms assumption and algorithms that require non-i.i.d. grouped data. We review the use of exchangeability in causality and approaches for causal structure learning in grouped data in Section 7. In Section 6, we present experiments that validate our approach to inferring causal structure from multi-environment data using conditional independence testing. Fig. 1 summarizes the main contributions of the paper and Fig. 2 illustrates differences between data generated by causal graph under i.i.d. process and ICM-generative process.

## 2 Preliminaries

### 2.1 The Causal Framework

A joint distribution $P(X_1, \ldots, X_N)$ over a set of variables $X_1, \ldots, X_N$ can be decomposed into simpler components in multiple ways. For example, by the chain rule of probability, one can factorize

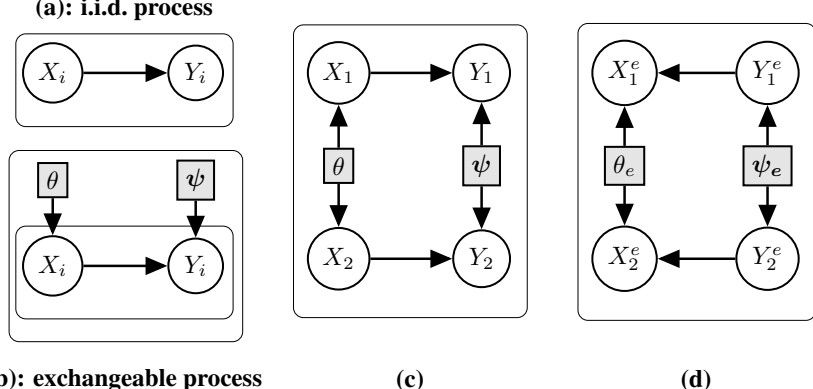

**(a): i.i.d. process**

**(b): exchangeable process**        **(c)**        **(d)**

Figure 2: An illustration demonstrates different conditional independence relationships contained in i.i.d. process and ICM-generative process. (a): A causal graph generated under an i.i.d. process; (b): A causal graph generated under ICM-generative process; Unrolling the inner plate notation from (b), we visualize the process with two samples. Causal graphs $X \to Y$ and $Y \to X$ generated under an i.i.d. process share the same conditional independences $\{\emptyset\}$ and are thus observationally equivalent. (c) and (d) show the corresponding graphs under ICM-generative processes. (c) has $X_1 \perp\!\!\!\perp Y_2 \mid X_2$ which does not hold in (d) and (d) has $X_1 \perp\!\!\!\perp Y_2 \mid Y_1$ which does not hold in (c). One can thus differentiate the bivariate causal direction in ICM-generative processes.

the joint as $P(X_1, \ldots, X_N) = \prod_{i=1}^{N} P(X_i \mid X_1, \ldots, X_{i-1})$. We say the joint distribution satisfies the *Markov factorization* property with respect to a directed acyclic graph $\mathcal{G}$ if

$$P(X_1, \ldots, X_N) = \prod_{i=1}^{N} P(X_i \mid \mathbf{PA}_i), \tag{1}$$

where $\mathbf{PA}_i$ are the direct parents of node $X_i$ in $\mathcal{G}$. While many factorisations can represent the same joint $P$, a specific one is called the *causal Markov factorization*: in it, the factors $P(X_i \mid \mathbf{PA}_i)$ represent the causal mechanisms of the true underlying data generating process. One can use the causal factorization to predict effects of interventions, which we model mathematically by replacing the corresponding factor (Pearl, 2009).

Causal discovery aims at recovering the causal graph $\mathcal{G}$ and the corresponding causal Markov factorization from the joint distribution $P$. This can be done by matching the conditional independence structure implied by the graph $\mathcal{G}$ to those observable in the joint distribution $P$. To facilitate this matching, an elaborate graphical terminology has been developed, as detailed in Appendix A. Unfortunately, under the assumption that data is sampled i.i.d. from $P$, the true underlying $\mathcal{G}$ cannot be uniquely determined, only up to broad equivalence classes. The conditional independence structure of i.i.d. processes is not rich enough to facilitate identifiability of the causal structure $\mathcal{G}$.

**Independence of Causal Mechanisms, Causal and Anti-Causal Machine Learning** In addition to the study of Markov factorization, recent work (Janzing and Schölkopf, 2010) studies the behaviour of causal mechanisms and postulates the *Independent Causal Mechanism (ICM)* principle, which states:

> Causal mechanisms are independent of each other in the sense that a change in one mechanism $P(X_i \mid \mathbf{PA}_i)$ does not inform or influence any of the other mechanisms $P(X_j \mid \mathbf{PA}_j)$, for $i \neq j$.

The notion of invariant, independent and autonomous mechanisms have a long history in causality research: Haavelmo (1944) and Aldrich (1989) discuss the historical development of autonomous mechanisms in economics and Pearl (2009) also argues that causal mechanisms are modular in nature. Schölkopf et al. (2012) pointed out the implications of this principle when using machine learning techniques in *causal or anti-causal learning* settings, i.e., when the task is to predict an effect from a cause or a cause from an effect, respectively. The ICM principle implies that semi-supervised learning is only successful in the anti-causal direction, while the predictor can be robustly applied to new domains if learning is in the causal direction. While these observations seem intuitively true, it is difficult to ground their meaning in the langauge of probability or information. As we will see, these difficulties can be resolved once we consider non-i.i.d. data generating processes.

## 2.2 Exchangeability

As we have seen, i.i.d. processes have a limitation that their conditional independence structure is not rich enough to support identifiability of the full causal graph. We thus turn our attention to a richer class of processes, exchangeable sequences.

**Definition 1** (Exchangeable sequence). *A finite sequence of random variables $X_1, X_2, \ldots, X_N$ is **exchangeable**, if for any permutation $\pi$ of its indices $\{1, \ldots, N\}$:*

$$P(X_{\pi(1)}, \ldots, X_{\pi(N)}) = P(X_1, \ldots, X_N) \tag{2}$$

*An **infinite exchangeable** sequence is a sequence where for any $N \in \mathbb{N}$, its finite sequence of length $N$ is exchangeable.*

Exchangeability is a notion of symmetry. Definition 1 informally states the order of observations does not matter. Recall a finite sequence of random variables is *independent and identically distributed* if its joint distribution satisfies $P(X_1, \ldots, X_N) = \prod_{i=1}^{N} P(X_i)$. Of course, such an i.i.d. sequence is automatically exchangeable but not all exchangeable sequences are i.i.d. To clarify the connection between exchangeable and i.i.d. sequences, recall the de Finetti theorem:

**Theorem 1** (De Finetti's representation theorem (de Finetti, 1931)). *Let $(X_n)_{n \in \mathbb{N}}$ be an infinite sequence of binary[*] random variables. The sequence is exchangeable if and only if there exists a random variable $\theta \in [0, 1]$ such that $X_1, X_2, \ldots$ are conditionally independent and identically distributed given $\theta$, with a probability measure $\mu$ on $\theta$. Mathematically speaking, given any sequence $(\mathbf{x}_1, .., \mathbf{x}_N) \in \{0, 1\}^N$:*

$$P(\mathbf{x}_1, \ldots, \mathbf{x}_N) = \int \prod_{i=1}^{N} p(\mathbf{x}_i \mid \theta) d\mu(\theta) \tag{3}$$

Informally, the theorem states that an exchangeable sequence can always be represented as a mixture of i.i.d. sequences. De Finetti's representation theorem has important consequences for Bayesian inference. Bayesian statistics, unlike frequentist, takes the view that the parameter is a latent variable, instead of an unknown point estimate. Bayes' theorem estimates the parameter via calculating posterior density $p(\theta | \mathbf{x}_1, .., \mathbf{x}_N)$. De Finetti's representation theorem demonstrates (O'Neill, 2009) that rather than metaphysical belief about the true model, it is due to our judgement that the observations are exchangeable that underlies our standard use of Bayesian modelling involving observations are i.i.d. conditioned on some unknown latent variable.

## 3 Causal de Finetti Theorems

Just as de Finetti justifies Bayesian modelling, we will introduce causal de Finetti theorems that justify causal modelling via probability theory. We first motivate our study of exchangeable sequences by noting that they have a richer conditional dependence structure than i.i.d. sequences. We illustrate what this means concretely in the simplest possible case of a pair of variables.

Let $X$ denote a random variable and $x$ denote a random variable's particular realization. Let $[n]$ denotes the set of positive integers that are less than or equal to $n$, i.e. $[n] = \{1, \ldots, n\}$.

**Definition 2** (Exchangeable pairs). *An infinite sequence of random variable pairs $(X_n, Y_n)_{n \in \mathbb{N}}$ is exchangeable if for any permutation $\pi$ and for any finite number $N$, it satisfies*

$$P(X_{\pi(1)}, Y_{\pi(1)}, \ldots, X_{\pi(N)}, Y_{\pi(N)}) = P(X_1, Y_1, \ldots, X_N, Y_N) \tag{4}$$

In an i.i.d. sequence over pairs, the only non-trivial independence is $X_i \perp\!\!\!\perp Y_i$. Since the distribution is identical, it either holds for all $i$ or does not hold for all $i$. In an exchangeable sequence of pairs, one can consider other non-trivial independence, for example, $Y_i \perp\!\!\!\perp X_j \mid X_i$. This conditional independence relationship trivially holds in i.i.d. sequences, but it may or may not hold in exchangeable sequences. Therefore, one can hope its absence or presence carries some useful information about some underlying causal structure. Here, we present the causal de Finetti theorems, which illustrate the type of causal structure this conditional independence relationship implies.

---

[*]De Finetti's representation theorem has been extended to categorical and continuous variables (Klenke, 2008).

**Theorem 2** (Causal de Finetti – bivariate)**.** *Let $\{(X_n, Y_n)\}_{n\in\mathbb{N}}$ be an infinite sequence of binary random variable pairs. The sequence is:*

1. *infinitely exchangeable, and satisfies*

2. $\forall n \in \mathbb{N}: Y_{[n]} \perp\!\!\!\perp X_{n+1} \mid X_{[n]}$

*if and only if there exist two random variables $\theta \in [0,1]$ and $\boldsymbol{\psi} \in [0,1]^2$ with probability measures $\mu, \nu$ such that the joint probability can be represented as*

$$P(x_1, y_1, \ldots, x_N, y_N) = \int \prod_{n=1}^{N} p(y_n \mid x_n, \boldsymbol{\psi}) p(x_n \mid \theta) d\mu(\theta) d\nu(\boldsymbol{\psi}) \tag{5}$$

Informally, the theorem states that an exchangeable sequence of random variable pairs satisfying an additional set of conditional independence properties, can always be represented as a mixture of i.i.d. sequences which all share the same underlying Markov factorization structure, and thus, an *invariant causal structure*. In fact, such exchangeable data can be interpreted as data generated under the ICM assumption. Recall the independent causal mechanism is loosely denoted as "$P_{\text{effect}|\text{cause}} \perp\!\!\!\perp P_{\text{cause}}$". ICM assumption can be modelled by Eq. 5 in the sense that causal mechanisms are characterized by latent variables: mechanisms do not influence each other if one can separately manipulate each latent variable controlling different mechanisms; further, latent variables governing each mechanism are statistically independent with each other, supporting mechanisms do not inform one another. We thus call the generative process in Eq. 5 as ICM-generative process. Causal de Finetti, just as how de Finetti substantiates Bayesian modelling, demonstrates that rather than metaphysical belief about independent causal mechanisms, it is due to our judgement that observations are exchangeable and the sufficiency to predict the target variable $Y$ with the corresponding $X$ values irrespective of other $X$ observations underlie our standard use of causal modelling. We thus substantiate ICM by detailing the statistical assumptions one implicitly make when assuming ICM.

As an example, consider the causal graph on the right. Imagine in a hospital there are two patients. A patient's symptom is the cause of a doctor's diagnosis. Suppose we are interested to predict a patient's diagnosis given her symptom. The conditional independence says knowing another patient's symptom will not help us to predict the diagnosis of this patient if we know this patient's symptoms already. The conditional independence thus formulated the intuition behind causal and anti-causal problem in the language of probability: the distribution of the cause, other patients' symptoms in this case, will not help prediction about the effect given cause, i.e., one patient's diagnosis given his own symptoms.

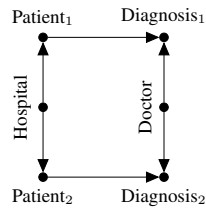

**Causal de Finetti vs. de Finetti** To see the difference between causal de Finetti and de Finetti's representation theorem, we observe that a direct application of de Finetti theorem on exchangeable data that may or may not contain causal information results in a factorization as:

$$P(x_1, y_1, \ldots, x_N, y_N) = \int \prod_{n=1}^{N} p(y_n, x_n \mid \theta) d\mu(\theta) \tag{6}$$

As observations are conditionally i.i.d. given latent variable $\theta$, learning $\theta$ is thus sufficient to achieve maximum prediction power. This finding corroborates empirical results in the machine learning community, where training often produces an entangled representation that achieves strong prediction accuracy. However, with the fast development of powerful machine learning applications (Brown et al., 2020), both deep learning and causal communities advocate the need for disentangled representations (Bengio et al., 2013; Locatello et al., 2019; Schölkopf et al., 2021), which offer greater control, interpretability, and generalization capabilities. Causal de Finetti shows that, in fact, given exchangeable data satisfying the causal and anti-causal learning phenomenon formulated in conditional independences, there are statistically independent latent variables controlling each causal mechanism. It shows one can achieve both maximum prediction power and disentangled representations. Fig. 1b illustrates a visualization of the differences between de Finetti and causal de Finetti theorems.

We next illustrate causal de Finetti in the general multivariate form (see Appendix B for proof):

**Definition 3** (Exchangeable arrays). *An array of size $d$ contains variables $(X_{1;n}, \ldots, X_{d;n})$ where $X_{d;n}$ denotes the $d$-th random variable in $n$-th array. Such an array is denoted as $\mathbf{X}_{:;n}$. A finite sequence of size $d$ arrays is **exchangeable**, if*

$$P(\mathbf{X}_{:;\pi(1)}, \ldots, \mathbf{X}_{:;\pi(N)}) = P(\mathbf{X}_{:;1}, \ldots, \mathbf{X}_{:;N}) \tag{7}$$

**Theorem 3** (Causal de Finetti – multivariate). *Let $\{(X_{1;n}, X_{2;n}, \ldots X_{d;n})\}_{n \in \mathbb{N}}$ be an infinite sequence of $d$-tuple binary random variables. The sequence is*

1. *infinitely exchangeable, and*

2. *if there exists a DAG $\mathcal{G}$ such that $\forall i \in [d], \forall n \in \mathbb{N}$:*

$$X_{i;[n]} \perp\!\!\!\perp \overline{\boldsymbol{ND}}_{i;[n]}, \boldsymbol{ND}_{i;n+1} | \boldsymbol{PA}_{i;[n]}$$

*where $\boldsymbol{PA}_i$ selects parents of node $i$ and $\boldsymbol{ND}_i$ selects non-descendants of node $i$ in $\mathcal{G}$. $\overline{\boldsymbol{ND}}_i$ denotes the set of non-descendants of node $i$ excluding its own parents.*

*if and only if there exist $d$ random variables where $\boldsymbol{\theta_i} \in [0,1]^{2^{|\boldsymbol{PA}_i|}}$ with suitable probability measures $\{\nu_i\}$ such that the joint probability can be written as*

$$P(\mathbf{x}_{:,1:N}) = \int \int \prod_{n=1}^{N} \prod_{i=1}^{d} p(x_{i;n} \mid \boldsymbol{pa}_{i;n}, \boldsymbol{\theta_i}) d\nu_1(\boldsymbol{\theta_1}) \ldots d\nu_d(\boldsymbol{\theta_d}), \tag{8}$$

*where $\mathbf{x}_{:,1:N} := \{(x_{1;n}, \ldots, x_{d;n})\}_{n=1}^{N}$.*

Theorem 2 is a special case of Theorem 3 when $d = 2$. Informally, it states that an exchangeable sequence of size $d$ random arrays satisfying an additional set of conditional independence properties with respect to a DAG, can always be represented as a mixture of i.i.d. sequences which all share the same underlying Markov factorization structure as the corresponding DAG. The set of conditional independence properties in condition 2 can be interpreted via its decomposition:

$$X_{i;[n]} \perp\!\!\!\perp \overline{\mathbf{ND}}_{i;[n]} \mid \mathbf{PA}_{i;[n]}$$

This shows that the direct parents of one node form a Markov blanket for its other non-descendant nodes. In other words, to infer about the variable of interest, it is sufficient to know the variable's direct parents irrespective of other non-descendants.

$$X_{i;[n]} \perp\!\!\!\perp \mathbf{ND}_{i;n+1} \mid \mathbf{PA}_{i;[n]}$$

This conditional independence is another example that exchangeable data has richer conditional independence structure. It trivially holds in i.i.d. sequences but may or may not hold in exchangeable data. Informally, it states that to infer the variable of interest, it is sufficient to know its corresponding parents irrespective of non-descendants in other observations. This formulated the intuition behind the causal and anti-causal problem in the language of probability: the distribution of the variables in the causal direction, in this case, non-descendants in other observations will not help prediction if covariates contain a complete set of the corresponding direct causal parents.

To illustrate how Theorem 3 also justifies the ICM in action, consider the causal graph to the right. Imagine high schools host campaigns to advertise university opportunities and encourage students to apply. Students' decision to apply is the cause of their university admission outcomes. Suppose we are interested in understanding how influential school campaigns are on students' decision to apply, i.e., the mechanism of "Apply | Campaign". We expect that knowing a particular student's decision to apply after attending a school campaign 
will not be influenced by the other school campaigns the student did not attend; instead, knowing other students' decisions to apply after their attendance in campaigns will help the prediction of this particular student's decision to apply. Similarly, knowing other students' university admission outcomes will also be helpful. This is because more students decide to apply implies the effectiveness of school campaigns, and more university admission acceptance implies more students decide to apply. Causal de Finetti says given such assumptions and exchangeable data, it naturally holds that there exist latent variables, represented by high school, student and university, and they are independent.

**Extension beyond binary** Above theorems are presented in its bivariate and multivariate forms for binary variables. In general, it is easy to extend the results to categorical variables[*]. Just as the progression of the proofs for de Finetti's theorem, we hypothesize causal de Finetti holds true for continuous variables. Here we state the theorem in its multivariate form for categorical variables.

**Theorem 4** (Causal de Finetti – multivariate and categorical). *Consider an infinite sequence of size-$d$ random arrays $\{(X_{1;n}, X_{2;n}, \ldots X_{d;n})\}_{n \in \mathbb{N}}$, where each variable $X_{i;n}$ takes values in $\{1, \ldots, k_i\}$. The sequence is:*

1. *infinitely exchangeable, and*

2. *if there exists a DAG $\mathcal{G}$ such that $\forall i \in [d], \forall n \in \mathbb{N}$:*

$$X_{i;[n]} \perp\!\!\!\perp \overline{\boldsymbol{ND}}_{i;[n]}, \boldsymbol{ND}_{i;n+1} | \boldsymbol{PA}_{i;[n]}$$

*where $\boldsymbol{PA}_i$ selects parents of node $i$ and $\boldsymbol{ND}_i$ selects non-descendants of node $i$ in $\mathcal{G}$. $\overline{\boldsymbol{ND}}_i$ denotes the set of non-descendants of node $i$ excluding its own parents.*

*if and only if there exist $d$ random variables where $\boldsymbol{\theta_i} \in [0,1]^{k_i \times \prod_{X_j \in \boldsymbol{PA}_i} k_j}$ and every column of $\boldsymbol{\theta}_i$ sum to 1 with suitable probability measures $\{\nu_i\}$ such that the joint probability can be written as*

$$P(\mathbf{x}_{:,1:N}) = \int \int \prod_{n=1}^{N} \prod_{i=1}^{d} p(x_{i;n} | \boldsymbol{pa}_{i;n}, \boldsymbol{\theta_i}) d\nu_1(\boldsymbol{\theta_1}) \ldots d\nu_d(\boldsymbol{\theta_d}), \tag{9}$$

*where $\mathbf{x}_{:,1:N} := \{(x_{1;n}, \ldots, x_{d;n})\}_{n=1}^{N}$.*

## 4  Identifiability Result

The causal de Finetti theorems show that exchangeable processes can be represented as ICM generative process when certain CI statements hold. However, such a representation is only really useful for causal discovery if it is unique - in other words we would like if only one such decomposition were possible for any given exchangeable process. This property is called *identifiability*. In this section we study the identifiability of ICM generative process. We start by introducing graphical terminology.

**Definition 4** (Acyclic directed mixed graph (ADMG) (Richardson, 2003)). *An acyclic directed mixed graph can contain two types of edges: directed '$\rightarrow$' or bi-directed '$\leftrightarrow$'. When an ADMG does not contain any bi-directed edge, it becomes a directed acyclic graph (DAG).*

**Definition 5** ($\mathcal{I}$-map (Koller and Friedman, 2009)). *Given $P$ is a distribution, $\mathcal{I}(P)$ denotes the set of conditional independence relationships of the form $X \perp\!\!\!\perp Y \mid Z$ that hold in $P$. Given $\mathcal{G}$ be a ADMG, $\mathcal{I}(\mathcal{G})$ denotes the set of conditional independence assumptions encoded in $\mathcal{G}$ which can be directly read-off via m-separation (Zhang and Spirtes, 2012). When $\mathcal{G}$ is a DAG, $\mathcal{I}(\mathcal{G})$ can be directly read-off via d-separation (Pearl, 1988).*

**Definition 6** (Global Markov property and faithfulness (Zhang and Spirtes, 2012)). *Given a ADMG $\mathcal{G}$ and a joint distribution $P$, the distribution satisfies global Markov property with respect to $\mathcal{G}$ if $\mathcal{I}(\mathcal{G}) \subseteq \mathcal{I}(P)$. Sometimes it is also called $P$ is **Markovian** with respect to $\mathcal{G}$. We say $P$ is **faithful** to $\mathcal{G}$ if $\mathcal{I}(P) \subseteq \mathcal{I}(\mathcal{G})$.*

Definition 7 defines the mapping from causal graphs generated under an i.i.d. process to those generated under an exchangeable process. The latent variables in causal de Finetti theorems can be represented by bi-directed edges in Definition 4.

**Definition 7** (ICM operator on a DAG). *Let $U$ be the space of all DAGs whose nodes represent $X_1, \ldots, X_d$. Let $V$ be the space of ADMGs whose nodes represent $\{(X_{i;n})\}$, where $i \in [d], n \in \mathbb{N}$. A mapping $F$ from $U$ to $V$ is an ICM operator if $F(\mathcal{G})$ satisfies:*

- *$F(\mathcal{G})$ restricted to the subset of vertices $\{X_{1;n}, \ldots, X_{d;n}\}$ is a DAG $\mathcal{G}$, for any $n \in \mathbb{N}$,*

- *$X_{i;n} \leftrightarrow X_{i;m}$ whenever $n \neq m$ for all $i \in [d]$*

---

[*](Barrett and Leifer, 2009) provides an alternative proof of conditional de Finetti in quantum theory for categorical variables.

- *there are no other edges other than stated above*

We denote the resulting ADMG as $ICM(\mathcal{G})$. Let $\boldsymbol{PA}_{i;n}^{\mathcal{G}}$ denote the parents of $X_{i;n}$ in $ICM(\mathcal{G})$ and similarly for $\boldsymbol{ND}_{i;n}^{\mathcal{G}}$ for corresponding non-descendants.

**Theorem 5** (Identifiability Theorem). *Consider the set of distributions that are both Markovian and faithful to $ICM(\mathcal{G})$, i.e., $\mathcal{E}(\mathcal{G}) := \{P : \mathcal{I}(P) = \mathcal{I}(ICM(\mathcal{G}))\}$. Then,*

$$\mathcal{E}(\mathcal{G}_1) = \mathcal{E}(\mathcal{G}_2) \text{ if and only if } \mathcal{G}_1 = \mathcal{G}_2 \tag{10}$$

The set of graphs $\{ICM(\mathcal{G})\}$ characterizes the set of causal graphs for data sampled from ICM-generative processes. Given any two causal structures $\mathcal{G}_1$ and $\mathcal{G}_2$ underlying data sampled from ICM-generative processes, we say they are Markov equivalent if $\mathcal{E}(\mathcal{G}_1) = \mathcal{E}(\mathcal{G}_2)$. Theorem 5 states given a distribution $P$ that is Markovian and faithful to $ICM(\mathcal{G})$, one can identify its unique causal structure as each graph induces unique conditional independences. See Appendix C for proof.

**Connection to i. i. d.** Causal de Finetti theorems though stated formally under exchangeable process, it automatically holds for data generated under an i.i.d. process. When observing i.i.d. data, the measures $\nu_i$ in (8) are Dirac measures, and the de Finetti parameters $\{\theta_i\}$ are deterministic, i.e., fixed across multiple samples generated from the process. The identifiability theorem stated here, however, excludes distributions generated by marginally i.i.d. processes. It requires $P$ to be faithful to $ICM(\mathcal{G})$. If any one of the marginal distributions of $P$ can collapse to an i.i.d. process, i.e., there exists an index $d$ such that $P(X_{d;1}, \ldots, X_{d;N}) = \prod_n P(X_{d;n})$, then $P$ is not faithful to $ICM(\mathcal{G})$ since it contains extra conditional independence relationships. Fig. 2 illustrates that compared to i. i. d process, ICM-generative processes enable unique causal structure identification.

## 5 Causal Structure Learning in Multi-environment Data

We established in Thm. 5 that causal structure is identifiable in ICM generative models by testing for CI relationships in exchangeable data. For example, if $Y_i \perp\!\!\!\perp X_j \mid X_i$ holds for an exchangeable pair $(X_n, Y_n)$, we conclude that $X \rightarrow Y$, i. e. $X$ causes $Y$. But how exactly does one test for this in data?

To test if a CI statement holds between a set of random variables, one typically requires multiple samples, that is i.i.d. copies of the variables in question. Similarly, to apply our identification results in practice, we need access to multiple i.i.d. copies of the exchangeable pair $(X_n, Y_n), n \in \mathbb{N}$ (see Def. 2). Each copy of $\{(X_n, Y_n)\}_{n \in \mathbb{N}}$ gives us a whole dataset containing a sequence of individual pairs, thus, we need multiple independent datasets to test for the CI condition. This requirement for multiple datasets connects our work to grouped or multi-environment data.

In the causal literature, grouped data refers to data available from multiple environments, each producing (conditionally) i. i. d observations from a different distribution, which are related through some *invariant causal structure* shared by all environments. Grouped data underlies a wide range of causal discovery approaches (Arjovsky et al., 2019; Heinze-Deml et al., 2018; Huang et al., 2020; Peters and Meinshausen, 2016; Rojas-Carulla et al., 2018; Tian and Pearl, 2001). We can interpret multi-environment data through the lens of exchangeability as follow: In each environment $e \in \mathcal{E}$, we observe exchangeable samples $\mathbf{X}_{:;1:N_e}^e = \{(X_{1;n}^e, \ldots, X_{d;n}^e)\}_{n=1 \ldots N_e}$, where $X_{d;n}^e$ denotes the $d$-th random variable in $n$-th sample in environment $e$ and $N_e$ is the number of samples from environment $e$. Data across enviroments are independent and identically distributed in the sense that the distribution of $\mathbf{X}_{:;1:N}^e$ and $\mathbf{X}_{:;1:N}^{e'}$ is identical for all $N < \min(N_e, N_{e'})$. Each environment thus provides a finite marginal of an i. i.d copy of the same exchangeable process, just as we needed for testing CI. Alternatively, one can also interpret environments as samples from latent variables, i.e. $(\theta^e, \psi^e)$ i.i.d. drawn from $p(\theta), p(\psi)$ characterizes environment $e$.

Next, we propose the *Causal-de-Finetti* algorithm, which guarantees to recover the DAG given multi-environment data consistent with ICM. In particular, the algorithm utilizes two samples per environment and a sufficiently large number of independent environments to enable identification.

**Notation**: As every sample in all environments shares the same causal structure, we sometimes abbreviate the variable $X_{i;n}^e$ to $X_{i;n}$ or $X_i$. The results proved under abbreviated indices mean the abbreviated indices could take any values and the result remains the same. Let $S_n$ denotes the set containing nodes belong to $n-$th rank in a DAG $\mathcal{G}$'s topological ordering. (See details in Appendix D)

**Lemma 1.** *A node $X_{i;n} \in S_1$ if and only if for every $m \neq n$ and $j \neq i$, $X_{i;n} \perp\!\!\!\perp X_{j;m} \mid \{X_{k;n}\}_{k \neq i}$.*

**Lemma 2.** *Let node $X_i \in S_n$ and $X_j \in S_m$ where $m < n$. Set $k := n - m$. There does not exist a directed edge from $X_i$ to $X_j$ if and only if when $k = 1$, $X_i \perp\!\!\!\perp X_j \mid S_{>n}$; and when $k > 1$: $X_i \perp\!\!\!\perp X_j \mid Z$, where $Z = S_{>n} \cup (\boldsymbol{PA}_j \cap S_{<n}) \cup (S_n \backslash X_i)$.*

Lemma 1 states the necessary and sufficient conditions to identify leaf nodes. Lemma 2, intuitively says, to decide whether $X_i$ and $X_j$ have a direct edge, one should block all the potential non-directed paths. Step 1 of the algorithm is to iteratively identify and remove the current set of leaf nodes, and then search for the next set of leaf nodes until all nodes have been classified into their topological orders. Step 2 of the algorithm is to apply Lemma 2 to determine the existence of an edge between different topological orders. Algorithm 1 in Appendix D details the exact procedure.

# 6 Experiments

We benchmark our method's performance against several state-of-the-art methods. As a measure of performance against methods for heterogeneous data, we compare against CD-NOD (Huang et al., 2017, 2020; Zhang et al., 2017). As a measure of performance against methods designed for i.i.d. data, we compare with common causal structure learning algorithms, e.g. FCI (Spirtes et al., 1995), GES (Chickering, 2002), NOTEARS (Zheng et al., 2018), DirectLiNGAM (Shimizu et al., 2011) and PC algorithm (Spirtes et al., 2000b). Lastly, we compare with a random guess baseline.

**Bivariate Causal Discovery** We generate multi-environment data as described in Section 5. Latent factors $\mathbf{N}$ were randomly generated with distinct and independent elements in each environment. Samples within each environment have the noise variables $\tilde{\mathbf{N}}$ generated via Laplace distributions conditioned on the latent factor. We observe bivariate data $\mathbf{X} \in \mathbb{R}^2$ with $X_1$ and $X_2$ denotes the first and second entry of $\mathbf{X}$ and aim to uncover the causal direction between $X_1$ and $X_2$. Let $^e$ denote variables contained in environment $e$.

$$\mathbf{N^e} \sim \mathcal{U}[-1, 1]$$
$$\tilde{\mathbf{N}}^{\mathbf{e}} \sim \text{Laplace}(\mathbf{N}, 1)$$
$$\mathbf{X^e} = \mathbf{A^e}\tilde{\mathbf{N}}^{\mathbf{e}} + \mathbf{B^e}\tilde{\mathbf{N}}^{\mathbf{e} \circ 2}\mathbb{1}_{\text{nonlinear}}(e)$$

where $\circ 2$ denotes elementwise square operation. Specifically, $\mathbf{A}^e \in \mathbb{R}^{2 \times 2}$ is a randomly sampled triangular matrix and $\mathbf{B^e} = \mathbf{A^e} - \mathbf{I}$. We randomly sample bivariate structures, $X_1 \rightarrow X_2, X_2 \rightarrow X_1, X_1 \perp\!\!\!\perp X_2$, by ensuring $\mathbf{A}$ is either a lower triangular, upper triangular or diagonal matrix. Our data further simulates a realistic situation, i.e., the causal structure remains invariant with changing functional relationships across environments. This is implemented by randomly sampling the existence of nonlinear dependence indicator $\mathbb{1}_{\text{nonlinear}}(e)$ per environment. We perform three conditional independence tests with $\alpha = 0.05$ and output our estimate as the causal structure corresponding to the test with the highest $p$-value. We repeat the experiment for 100 times and report the proportion of correct causal direction identified with varying numbers of environments. Figure 3(a) shows the proportion of correct bivariate causal direction detected as the number of environments $|\mathcal{E}|$ increases and the number of observations within each environment as fixed to be 2. We observe Causal-de-Finetti algorithm outperforms all the other state-of-the-art methods and its accuracy converges close to 100%. This demonstrates its capability to handle datasets with limited samples per environment and changing functional relationships across environments.

**Multivariate Causal Structure Discovery** We further test our algorithm's performance in identifying multivariate causal structure. We randomly generate causal graphs with 3 variable nodes where each variable takes binary values. Figure 3(b) shows the Structural Hamming Distance (Tsamardinos et al., 2006) between true and estimated DAG averaged over 100 experiments. We again observe Causal-de-Finetti outperforms all the other baselines, which validates our identifiability theorem and demonstrates algorithms designed for i.i.d. data does not work in our setting.

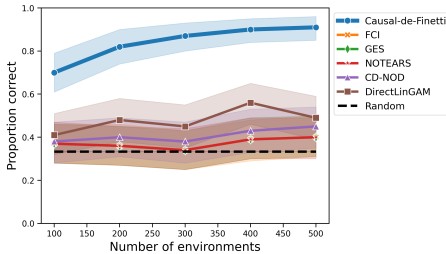
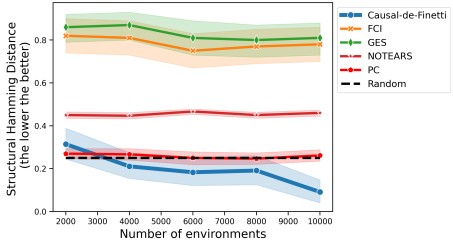

(a) Bivariate causal discovery    (b) Multivariate structure learning

Figure 3: Our method's ("Causal-de-Finetti") performance in identifying the correct underlying DAG, compared to the "CD-NOD", "FCI", "GES", "NOTEARS", "DirectLinGAM", "PC", "Random" baseline in bivariate and multivariate settings. Shown are the mean and $95\%$ confidence interval of the standard error of the mean for each method aggregated over 100 experiments. "Causal-de-Finetti" identifies unique causal structures and is robust against changing functions across environments.

## 7   Discussion

**Causal exchangeability** Dawid (2021) introduces a decision-theoretic framework for causality and uses pre-treatment and post-treatment exchangeability as foundational assumptions on external data used to solve the decision problem. Jensen et al. (2020) studies object conditioning and show its probabilistic interpretations can be explained using exchangeability. Object conditioning, due to exchangeability, thus mitigates latent confounding and measurement errors for causal inference. Our work provides a statistical understanding of ICM assumption: it is equivalent to assuming exchangeability and certain conditional independence conditions.

**Causal structure learning** Within the study of i.i.d. data, it is well-known that one can only identify causal structures up to Markov equivalence classes (Pearl, 1988), and going beyond that is known to be impossible without further parametric constraints (Hoyer et al., 2008b; Shimizu et al., 2006). A recent line of work considers a mixture of observational multi-environment data and interventional data to perform inference: Arjovsky et al. (2019); Peters and Meinshausen (2016); Rojas-Carulla et al. (2018) use causal invariance property to discover stable predictors and Huang et al. (2017, 2020) estimate kernel mean embeddings of heterogenous data distributions to test the independence of causal mechanisms. Monti et al. (2020) discovers bivariate causal direction by first recovering the underlying generating sources and then performing conditional independence tests on the recovered source factors. These algorithms on non-i.i.d. grouped data all demonstrate success, though it is unclear the connection between causal assumptions and probabilistic implications of grouped data. Our work observes that grouped data is akin to exchangeable sequences, containing richer conditional independence structures. In particular, ICM-generative processes with a sufficient number of environments allow unique causal structure identification.

**Relations to causality in time-series** The study of data generated from i.i.d. process, exchangeable process, and time-series can be seen as a progression to understand more structured data. Appendix E details the connections between ICM-generative processes and causality in time series (Runge, 2020).

**Conclusion** We prove causal de Finetti theorems formalizing the independent causal mechanism assumption in data generating processes as concrete statistical conditions. We call the induced generative models ICM-generative processes. For data sampled from ICM-generative processes, we show that one can identify unique causal structure. We build the connection between exchangeable and grouped data and justify the success of many methods leveraging ICM and algorithms in non-i.i.d. grouped data. Going beyond the i.i.d. assumption has been the bottleneck to applying machine learning to real-world situations. Rather than considering it a nuisance, our work shows an example of a theoretical advantage of exchangeable data in causal structure identification.

**Acknowledgement** S.G. would like to acknowledge helpful discussions with Damon Wishick on understanding causal de Finetti theorem in multi-environment data and Thijs van Ommen for his constructive feedback in the reviewing process which corrected the original version of Lemma 2.

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
