# A   Graphical Terminology

An arbitrary graph $\mathcal{G}$ consists of vertices $V$ and edges $E \subseteq V^2$ with $(v,v) \notin E$ for any $v \in V$. Then $\mathcal{G} = (V, E)$ is a graph with $V := \{1, \ldots, d\}$ and corresponding random variables $\{X_1, \ldots, X_d\}$. A variable $X_i$ is called a parent of $X_j$ if $(i,j) \in E$ and $(j,i) \notin E$ and a child if $(j,i) \in E$ and $(i,j) \notin E$. The set of parents of $X_j$ in $\mathcal{G}$ is denoted as $\mathbf{PA}_i^{\mathcal{G}}$, and the set of its children by $\mathbf{CH}_i^{\mathcal{G}}$. Whenever the graph $\mathcal{G}$ is obvious from the context, one can omit its specification in the above notations. Two variables $X_i$ and $X_j$ are adjacent if either $(i,j) \in E$ or $(j,i) \in E$. A pair of variables can be connected with a directed edge $X_i \rightarrow X_j$. If there does not exist a sequence of edges such that $X_i \rightarrow \cdots \rightarrow X_i$ for all $i \in V$, then $\mathcal{G}$ is acyclic.

A path in $\mathcal{G}$ is a sequence of (at least two) distinct vertices $i_1, \ldots, i_m$ such that there is an edge between $i_k$ and $i_{k+1}$ for all $k = 1, \ldots, m-1$. If $i_k \rightarrow i_{k+1}$ for all $k$, then $X_{i_1}$ is an ancestor of $X_{i_m}$, and that $X_{i_m}$ is a descendant of $X_{i_1}$. The set of ancestors of $X_i$ is denoted as $\mathbf{AN}_i^{\mathcal{G}}$ and $\mathbf{DE}_i^{\mathcal{G}}$ denotes the set of descendants of $X_i$. All non-descendants of $X_i$, excluding itself, are denoted as $\mathbf{ND}_i^{\mathcal{G}}$. In this work we use $\overline{\mathbf{ND}}_i^{\mathcal{G}}$ to denote the set of non-descendants excluding its parents.

Causal structure learning via performing conditional independence tests involves matching conditional independences contained in probability distributions with the conditional independence assumptions encoded in the graph. D-separation (Pearl, 1988) provides a graphical criterion that characterizes the set of conditional independences in the graph.

**Definition 8** (d-separation). *Given a directed acyclic graph $\mathcal{G}$, a path $p$ with vertices $i_1, \ldots, i_m$ is d-separated by a block of nodes $Z$ if and only if one of the two conditions holds:*

- *$p$ contains a chain $i_{k-1} \rightarrow i_k \rightarrow i_{k+1}$ or $i_{k-1} \leftarrow i_k \leftarrow i_{k+1}$, or a fork $i_{k-1} \leftarrow i_l \rightarrow i_{k+1}$ and $i_k \in Z$;*

- *$p$ contains a collider $i_{k-1} \rightarrow i_k \leftarrow i_{k+1}$ s.t. the middle node $i_k \notin Z$ and none of its descendants is in $Z$.*

*We then say $Z$ d-separates two disjoint subsets of vertices $X$ and $Y$ if it blocks every path from a node in $X$ to a node in $Y$, and write as $X \perp\!\!\!\perp_{\mathcal{G}} Y | S$.*

We refer the readers to (Peters et al., 2017) for more detailed graphical terminology.

# B   Proof of Causal de Finetti

Here we refer to Causal de Finetti as in its multivariate form, as bivariate is a subcase contained in multivariate form. We base our proof mostly on (Kirsch, 2019).

**Preliminaries** For a probability measure $\mu$ on $\mathbb{R}^d$ we define the mixed moment by $m_{\mathbf{a}}(\mu) := \int \prod_{i=1}^d x_i^{a_i} d\mu(x_1, \ldots, x_d)$ whenever it exists (in the sense that $\int \prod_{i=1}^d |x_i|^{a_i} d\mu(x_1, \ldots, x_d) < \infty$). Below we will only deal with measures with compact support so that all moments exist and are finite. The following is a multivariate extension of method of moments:

**Theorem 6** (Multivariate method of moments). *Let $\mu_n (n \in \mathbb{N})$ be probability measures with support contained in a fixed interval $[a,b]^d$. If for all $\mathbf{u}$ the mixed moments $m_{\mathbf{u}}(\mu_n)$ converge to some $m_{\mathbf{u}}$ then the sequence $\mu_n$ converges weakly to a measure $\mu$ with moments $m_{\mathbf{u}}(\mu) = m_{\mathbf{u}}$ and with support contained in $[a,b]^d$. Further, if $\mu$ is a probability measure with support contained in $[a,b]^d$ and $\nu$ is a probability measure on $\mathbb{R}^d$ such that $m_{\mathbf{u}}(\mu) = m_{\mathbf{u}}(\nu)$ then $\mu = \nu$.*

*Proof of Theorem 6.* The first statement follows directly from the first theorem in (Haviland, 1936). The second statement can be shown using a similar argument as univariate case in (Kirsch, 2019) with Weierstrass approximation theorem. $\square$

**Notation** Let $X_{i;n}$ denotes $i$-th random variable in $n$-th sample. We write $\mathbf{X_n} := (X_{1;n}, \ldots, X_{d;n})$ and $X_{d;:} := (X_{d;1}, \ldots, X_{d;N})$. Define $UP_{i;:} := (X_{i+1;:}, \ldots, X_{d;:})$, which contains all random variables that have higher variable index value than $i$, i.e. upstream of node $i$.

**Definition 9** (Topological Ordering). *A topological ordering of a DAG is a linear ordering of its nodes such that for every directed edge $X \to Y$, $X$ comes before $Y$ in the ordering. We call the ordering is a reversed topological order if we reverse the topological ordering of a DAG.*

Without loss of generality, we reorder the variables according to reversed topological ordering, i.e. a node's parents will be placed after this node. Note a reversed topological ordering is not unique, but it must satisfy a node's descendants will come before itself. Then by Kolmogorov's chain rule, we can always write the joint probability distribution as

$$P(\mathbf{X_1}, \ldots, \mathbf{X_N}) = \prod_{i=1}^{d} \tilde{P}(X_{i;:} | UP_{i;:}) \tag{11}$$

Note $P$ and $\tilde{P}$ are not the same, for ease of notation, we use $P$ below in general. For each $X_{i;:}$, we want to show there exists a suitable probability measure $\nu_i$ such that we can write $P(X_{i;:} | UP_{i;:}) = \int \prod_{n=1}^{N} p(X_{i;n} | PA_{i;n}, \boldsymbol{\theta_i}) d\nu_i(\boldsymbol{\theta_i})$. Then substitute back into Equation 11 we will have Causal de Finetti.

**Theorem 7** (Causal Conditional de Finetti). *Let $\{(X_{i;n}, X_{i+1;n}, \ldots X_{d;n})\}_{n \in \mathbb{N}}$ satisfies conditions 1) and 2) in Causal de Finetti. Then there exists a suitable probability measure $\nu$ such that the conditional probability can be written as*

$$P(X_{i;:} | UP_{i;:}) = \int \prod_{n=1}^{N} p(X_{i;n} | PA_{i;n}, \boldsymbol{\theta}) d\nu(\boldsymbol{\theta}) \tag{12}$$

*where $\boldsymbol{\theta}$ is a vector where its index represents a unique combination for $PA_i$ values. We can thus consider $p(X_{i;n} | PA_{i;n}, \boldsymbol{\theta}) = \pi_{\theta_j}(X_{i;n})$ where $j$ is the index of $PA_{i;n}$ in all possible realizations of $PA_i$ and $\pi_p$ is a Bernoulli probability measure parameterized by $p \in [0, 1]$.*

**Lemma 3.** *Let $\{(X_{i;n}, X_{i+1;n}, \ldots X_{d;n})\}_{n \in \mathbb{N}}$ satisfies conditions 1) and 2) in Causal de Finetti. Then for every permutation $\pi$ of $\{1, 2, ..., N\}$:*

$$\begin{aligned} &\mathbb{P}(X_{i;1}, ..., X_{i;N} | UP_{i,1}, ..., UP_{i;N}) \\ =&\mathbb{P}(X_{i;\pi(1)}, ..., X_{i;\pi(N)} | UP_{i;\pi(1)}, ..., UP_{i;\pi(N)}) \end{aligned} \tag{13}$$

*Proof of Lemma 3.* Since $\{(X_{i;n}, X_{i+1;n}, \ldots X_{d;n})\}_{n \in \mathbb{N}}$ is exchangeable, it is clear by marginalizing $\{X_{i;n}\}_{n \in \mathbb{N}}$ from definition we have $\{UP_{i;n}\}_{n \in \mathbb{N}}$ is also exchangeable.

$$\begin{aligned} &\mathbb{P}(X_{i;1}, ..., X_{i;N} | UP_{i,1}, ..., UP_{i;N}) \\ =&\frac{\mathbb{P}(X_{i;1}, UP_{i,1}, ..., X_{i;N}, UP_{i;N})}{\mathbb{P}(UP_{i,1}, ..., UP_{i;N})} \\ =&\frac{\mathbb{P}(X_{i;\pi(1)}, UP_{i;\pi(1)}, ..., X_{i;\pi(N)}, UP_{i;\pi(N)})}{\mathbb{P}(UP_{i;\pi(1)}, ..., UP_{i;\pi(N)})} \\ =&\mathbb{P}(X_{i;\pi(1)}, ..., X_{i;\pi(N)} | UP_{i;\pi(1)}, ..., UP_{i;\pi(N)}) \end{aligned}$$

Note since we reorder the index of multivariate random variables according to reversed topological ordering, we have $PA_{i;n} \subseteq UP_{i;n}$, so given $UP_{i;n}$ we would know $PA_{i;n}$. This lemma implies for every conditional distribution we can always choose a permutation such that we can group values with identical $PA_i$'s realizations together. For example, when $|PA_i| = 1$, then we can permute such that all observations with $PA_{i;n} = 0$ comes first and observations with $PA_{i;n} = 1$ come second. Let's order all possible realizations of $PA_i$ into a list of length $K := 2^{|PA_i|}$ and index each realization. Then from observations we have $N_k$ pairs which have $PA_i$ takes values as the index $k$'s realization. Here we assume we observe enough samples to see every realization of $PA_i$. This is possible because $K$ is finite. Then we can rearrange such that

$$\begin{aligned} &\mathbb{P}(X_{i;1}, ..., X_{i;N} | UP_{i,1}, ..., UP_{i;N}) \\ =&P(\{\{X_{i;n}^k\}_{n=1}^{N_k}\}_{k=1}^{K} | \{\{UP_{i;n}^k\}_{n=1}^{N_k}\}_{k=1}^{K}) \end{aligned}$$

where $X_{i;n}^k$ denotes that its parents $PA_{i;n}$ takes realizations the same as index $k$ indicates and $UP_{i;n}^k$ denotes that the random vector $UP_{i;n}$ contains $PA_i$ which takes realizations the same as index $k$ indicates. □

**Corollary 1.** *For any $K$-tuple permutations $(\pi_1, \pi_2, \ldots, \pi_K)$ where $\pi_k$ permutes $\{1, ..., N_k\}$:*

$$P(\{\{X_{i;n}^k\}_{n=1}^{N_k}\}_{k=1}^K | \{\{UP_{i;n}^k\}_{n=1}^{N_k}\}_{k=1}^K)$$
$$= P(\{\{X_{i;\pi_k(n)}^k\}_{n=1}^{N_k}\}_{k=1}^K | \{\{UP_{i;\pi_k(n)}^k\}_{n=1}^{N_k}\}_{k=1}^K)$$

*Proof of Corollary 1.* Follows directly from Lemma 1. $\qquad\square$

**Lemma 4.** *Recall condition 2) in Causal de Finetti states that $\forall i, \forall n \in \mathbb{N}$:*

$$X_{i;[n]} \perp\!\!\!\perp \overline{ND}_{i;[n]}, ND_{i;n+1} | PA_{i;[n]}$$

*By exchangeability, it is equivalent to*

$$X_{i;I} \perp\!\!\!\perp \overline{ND}_{i;I}, ND_{i;m} | PA_{i;I}$$

*where $I$ is any set and $m \notin I$.*

**Lemma 5.** *Let $\{(X_{i;n}, X_{i+1;n}, \ldots, X_{d;n})\}_{n \in \mathbb{N}}$ satisfies conditions 1) and 2) in Causal de Finetti. There exists $K$-infinitely exchangeable sequence $\{\{X_{i;n}^{k,*}\}_{n \in \mathbb{N}}\}_{k=1}^K$ such that for every $N_k \in \mathbb{N}, \forall k$ we have:*

$$P(\{\{X_{i;n}^{k,*}\}_{n=1}^{N_k}\}_{k=1}^K) \tag{14}$$
$$= P(\{\{X_{i;n}^k\}_{n=1}^{N_k}\}_{k=1}^K | \{\{UP_{i;n}^k\}_{n=1}^{N_k}\}_{k=1}^K)$$

*where $k$ is the index for $PA_i$'s particular realization.*

*Proof of Lemma 5.* To show such sequence exists, we need to show it is well-defined and the inductive defining sequence is consistent.
To show it is well-defined:

$$P(\{\{X_{i;n}^k\}_{n=1}^{N_k}\}_{k=1}^K | \{\{UP_{i;n}^k\}\}_{n=1}^{N_k}\}_{k=1}^K)$$
$$= P(\{\{X_{i;n}^k\}_{n=1}^{N_k}\}_{k=1}^K | \{\{PA_{i;n}^k\}\}_{n=1}^{N_k}\}_{k=1}^K)$$
$$= P(\{\{X_{i;n}^{k,*}\}_{n=1}^{N_k}\}_{k=1}^K)$$

Using Lemma 4, let $I = \{\{(k;n)\}_{n=1}^{N_k}\}_{k=1}^K$ and we have $\overline{UP}_i \subseteq \overline{ND}_i$ since any particular reversed topological sort will place node $i$'s decendants before itself. Then condition 2) implies $X_{i;I} \perp\!\!\!\perp \overline{UP}_{i;I} | PA_{i;I}$ by decomposition rule in conditional independence. Because index $k$ already characterizes the value of $PA_i$ so result follows by definition.
consistent: We write $\{\{\cdot\}_{n=1}^{N_k}\}_{k=1}^K$ as $\{\{\cdot\}\}$ for abbreviation. For any $k$, consider

$$P(\{\{X_{i;n}^{k,*}\}\})$$
$$= P(\{\{X_{i;n}^k\}\} | \{\{UP_{i;n}^k\}\})$$
$$= P(\{\{X_{i;n}^k\}\} | \{\{UP_{i;n}^k\}\}, UP_{i;N_k+1}^k)$$
$$= \sum_{X_{k;N_k+1}^i} P(\{\{X_{i;n}^k\}\}, X_{i;N_k+1}^k | \{\{UP_{i;n}^k\}\}, UP_{i;N_k+1}^k)$$
$$= \sum_{X_{k;N_k+1}^{i,*}=0}^1 P(\{\{X_{i;n}^{k,*}\}\}, X_{i;N_k+1}^{k,*})$$

The first equality holds by well-defindedness. Let $I = \{\{(k;n)\}_{n=1}^{N_k}\}_{k=1}^K$ and $m = (k; N_k + 1)$. Note $\overline{UP}_i \subseteq \overline{ND}_i$. Lemma 4 implies the second equality holds. The third equality holds by marginal property of probability distribution. The fourth equality follow from well-definedness. Infinite exchangeability of $\{X_{i;n}^{k,*}\}_{n \in \mathbb{N}}, \forall k$ follows from Corollary 1. $\qquad\square$

**Definition 10** (Causal Conditional de Finetti measure). *Using the notation introduced in Lemma 5, we define a random vector $\boldsymbol{Q}$ where $Q_k = \frac{1}{N_k} \sum_{n=1}^{N_k} X_{i;n}^{k,*}$. Let the joint distribution of $\boldsymbol{Q}$ be $\nu_{N_1;\ldots;N_K}$ or in shorthand $\nu_{\boldsymbol{N}}$ where $\boldsymbol{N} := [N_1, ..., N_K]$. If $\nu_{\boldsymbol{N}}$ converges to a probability measure $\nu$ as $N_k \to \infty, \forall k$, we call $\nu$ the Causal conditional de Finetti measure.*

**Lemma 6.** *Let $\{A_i\}_{i\in\mathbb{N}}$ be an infinitely exchangeable random binary process. Given a list of indices $\{i_1,\ldots,i_n\}$, let us denote the number of unique elements with $\rho(i_1,\ldots,i_n)$. For every arbitrary list of indices, the following holds:*

$$\mathbb{E}[A_{i_1}A_{i_2}\ldots A_{i_n}] = \mathbb{E}[A_1 A_2 \ldots A_{\rho(i_1,\ldots,i_n)}] \tag{15}$$

*Proof of Lemma 6.* For binary variables, we have $\mathbb{E}[A_i^l] = \mathbb{E}[A_i], \forall l \geq 1$. So the product from left hand side is a product of $\rho(i_1,\ldots,i_n)$ different $A_i$'s. Due to exchangeability, we have right hand side. $\square$

**Lemma 7.** *Let $\{\{A_{k;i}\}_{i\in\mathbb{N}}\}_{k=1}^{K}$ be $K$ infinitely joint exchangeable random binary processes. For every $K$ arbitrary list of indices $\{\{\mathbf{i_k}\}\}_{k=1}^{K}$, where $\mathbf{i_k} := (i_{k;1},\ldots,i_{k;N_k})$ denotes the sequence of indices selected for $k$-th process and $N_k$ is the number of indices. the following holds:*

$$
\begin{aligned}
&\mathbb{E}[\prod_{k=1}^{K}\prod_{i\in\mathbf{i_k}} A_{k;i}] \\
=&\mathbb{E}[\prod_{k=1}^{K}\prod_{i=1}^{\rho(\mathbf{i_k})} A_{k;i}]
\end{aligned}
\tag{16}
$$

*Proof.* Since the two sequences are jointly exchangeable, for $K$ sets of indices, we can independently perform the argument in Lemma 6. Hence only the number of unique indices in each set matters, which is the same on both sides. $\square$

**Theorem 8.** *If we allow $N_k \to \infty, \forall k$, the probability measure $\nu_{\mathbf{N}}$ converges to a probability distribution $\nu$. The measure $\nu$ has the following joint moments:*

$$m_{\boldsymbol{u}}(\nu) = \mathbb{E}[\prod_{k=1}^{K}\prod_{n=1}^{u_k} X_{i;n}^{k,*}]$$

*Proof.* We will first examine the mixed moments of $\nu_{\mathbf{N}}$.

$$
\begin{aligned}
&\lim_{\mathbf{N}\to\infty} m_{\boldsymbol{u}}(\nu_{\mathbf{N}}) \\
=\ &\mathbb{E}\Big[\lim_{\mathbf{N}\to\infty}\Big(\prod_{k=1}^{K}\frac{1}{N_k^{u_k}}\Big(\sum_{n=1}^{N_k} X_{i;n}^{k,*}\Big)^{u_k}\Big)\Big] \\
=\ &\mathbb{E}\Big[\lim_{N_k\to\infty}\frac{1}{N_k^{u_k}}\Big(\sum_{\mathbf{i_k}}\prod_{n\in\mathbf{i_k}} X_{i;n}^{k,*}\Big) \\
&\quad \prod_{j\neq k}\lim_{N_j\to\infty}\Big(\frac{1}{N_j^{u_j}}\Big(\sum_{n=1}^{N_j} X_{j;n}^{i,*}\Big)^{u_j}\Big)\Big]
\end{aligned}
$$

The second equality is possible by Lebesgue dominated convergence theorem and we have $|Q_k| \leq 1, \forall k, N_k$. The third equality is due to the product of limits is the limit of products. Next for any $k$, we focus on understanding each individual limit.

$$
\begin{aligned}
&\lim_{N_k\to\infty}\Big(\frac{1}{N_k^{u_k}}\Big(\sum_{\mathbf{i_k}}\prod_{n\in\mathbf{i_k}} X_{i;n}^{k,*}\Big)\Big) \\
=\ &\lim_{N_k\to\infty}\frac{1}{N_k^{u_k}}\Big(\sum_{\mathbf{i_k}:\rho(\mathbf{i_k})<u_k}\prod_{n\in\mathbf{i_k}} X_{i;n}^{k,*}\Big) \\
&+\lim_{N_k\to\infty}\frac{1}{N_k^{u_k}}\Big(\sum_{\mathbf{i_k}:\rho(\mathbf{i_k})=u_k}\prod_{n\in\mathbf{i_k}} X_{i;n}^{k,*}\Big)
\end{aligned}
$$

When $N_k \to \infty$, since $X_{i;n}^{k,*}$ are binary variable, the first term becomes:

$$0 \leq \frac{1}{N_k^{u_k}} \sum_{\mathbf{i_k}:\rho(\mathbf{i_k})<u_k} \left( \prod_{n \in \mathbf{i_k}} X_{i;n}^{k,*} \right)$$
$$\leq \frac{1}{N_k^{u_k}} \sum_{\mathbf{i_k}:\rho(\mathbf{i_k})<u_k} 1 \tag{17}$$

The number of possible tuples of indices $\mathbf{i_k}$ with $\rho(\mathbf{i_k}) < u_k$ is at most $(u_k - 1)^{u_k} N_k^{u_k-1}$. Because we have $N_k^{u_k-1}$ possibilities to choose the possible candidates for $(i_{k;1}, \ldots, i_{k;N_k})$ as long as we fix the last remaining index to one of the indices we have already chosen, we will still satisfy $\rho(\mathbf{i_k}) < u_k$. Then for each of the $u_k$ positions in the $u_k - tuple$ we may choose one out of $u_k - 1$ candidates that we have chosen which gives $(u_k - 1)^{u_k}$ possibilities. This covers also tuples with less than $u_k - 1$ different indices as some of the candidates may not appear in the final tuple. Therefore, if $N_k \to \infty$, the expectation in Equation 17 to 0. Also note the number of possible tuples of indices $\mathbf{i_k}$ with $\rho(\mathbf{i_k}) = u_k$ is $\binom{N_k}{u_k}$. Hence the moment converges to:

$$\lim_{\mathbf{N} \to \infty} m_{\mathbf{u}}(\nu_{\mathbf{N}}) = \mathbb{E}\left[ \prod_{k=1}^{K} \prod_{n=1}^{u_k} X_{i;n}^{k,*} \right]$$

The equality follows from Lemma 7 and we know $\lim_{N_k \to \infty} \frac{\binom{N_k}{u_k}}{N_k^{u_k}} = constant, \forall k$. Without loss of generality, consider the constant to be 1, the remaining argument will not change. Using Theorem 6, we have there exists a probability measure $\nu$ such that $m_{\mathbf{u}}(\nu) = \mathbb{E}\left[ \prod_{k=1}^{K} \prod_{n=1}^{u_k} X_{i;n}^{k,*} \right]$ □

**Lemma 8.** *Let $\{\{A_{k;i}\}_{i \in \mathbb{N}}\}_{k=1}^{K}$ be $K$ infinitely joint exchangeable random binary processes. For any $K$ binary sequence $\{a_{k;1}, \ldots, a_{k;N_k}\}_{k=1}^{K}$ with $\sum_{n=1}^{N_k} a_{k;n} = r_k$:*

$$\mathbb{P}(\{\{A_{k;n} = a_{k;n}\}_{n=1}^{N_k}\}_{k=1}^{K})$$
$$= \frac{1}{\prod_{k=1}^{K} \binom{N_k}{r_k}} \mathbb{P}\left( \sum_{n=1}^{N_1} A_{1;n} = r_1, \ldots, \sum_{n=1}^{N_K} A_{K;n} = r_K \right)$$

*Proof of Lemma 8.* We can distribute the 1's for each sequence in $\prod_{k=1}^{K} \binom{N_k}{r_k}$ different ways. Due to $K$ sequences being exchangeable, all of them have the same probability. □

Next, we will prove Causal conditional de Finetti.

*Proof of Theorem 7.* Let $\nu$ be Causal conditional de Finetti measure for $\{\{X_{i;n}^{k,*}\}\}$ (see Definition 10) and define $K$ random binary processes $\{\{Z_n^k\}_{n \in \mathbb{N}}\}_{k=1}^{K}$ with the following finite dimensional distribution:

$$\mathbb{P}(\{\{Z_n^k = z_n^k\}_{n=1}^{N_k}\}_{k=1}^{K})$$
$$= \int \prod_{k=1}^{K} \prod_{n=1}^{N_k} \pi_{\theta_k}(z_n^k) d\nu(\boldsymbol{\theta})$$

Note from the definition, the series $\{\{Z_n^k\}_{n \in \mathbb{N}}\}_{k=1}^{K}$ is infinitely joint exchangeable. We will prove $\{\{X_{k;n}^{i,*}\}_{n \in \mathbb{N}}\}_{k=1}^{K}$ and $\{\{Z_n^k\}_{n \in \mathbb{N}}\}_{k=1}^{K}$ have the same finite dimensional distribution. Define the following random vector $\boldsymbol{R}$ where $R_k = \frac{1}{N_k} \sum_{n=1}^{N_k} Z_n^k$. By Lemma 8, it suffices to show that $\mathbf{Q}$ (as in Definition 10) and $\mathbf{R}$ have the same distributions for all $N_k \in \mathbb{N}$ and for all $k$ and by the second statment in Theorem 6, we know two probability distributions are identical if their moments agree.

$$\mathbb{E}\Big[\prod_{k=1}^{K}(Q_k)^{u_k}\Big] = \frac{1}{\prod_{k=1}^{K} N_k^{u_k}}\mathbb{E}\Bigg[\prod_{k=1}^{K}\Big(\sum_{n=1}^{N_k} X_{i;n}^{k,*}\Big)^{u_k}\Bigg]$$

$$= \frac{1}{\prod_{k=1}^{K} N_k^{u_k}}\mathbb{E}\Bigg[\prod_{k=1}^{K}\Big(\sum_{\mathbf{i_k}}\prod_{n\in\mathbf{i_k}} X_{i;n}^{k,*}\Big)\Bigg] \tag{18}$$

$$= \frac{1}{\prod_{k=1}^{K} N_k^{u_k}}\sum_{a_1=1}^{u_1}\cdots\sum_{a_K=1}^{u_K}\sum_{\substack{\forall k,\mathbf{i_k}:\\ \rho(\mathbf{i_k})=a_k}}\mathbb{E}\Big[\prod_{k=1}^{K}\prod_{n=1}^{a_k} X_{i;n}^{k,*}\Big]$$

The last equality follows from Lemma 7. From Theorem 8, we know the above expectations are in fact the moments of the probability measure $\nu$:

$$\mathbb{E}\Big[\prod_{k=1}^{K}\prod_{n=1}^{a_k} X_{i;n}^{k,*}\Big] = m_{\boldsymbol{a}}(\nu)$$

$$= \int\prod_{k}(\theta_k)^{a_k}d\nu(\boldsymbol{\theta})$$

$$= \int\prod_{k=1}^{K}\prod_{n=1}^{a_k}\pi_{\theta_k}(x_{i;n}^{k,*})d\nu(\boldsymbol{\theta})$$

$$= \mathbb{E}\Big[\prod_{k=1}^{K}\prod_{n=1}^{a_k} Z_n^k\Big]$$

Hence continuing Equation 18 and reverting the steps taken in Equation 18 and using Lemma 7 we have:

$$(18) = \frac{1}{\prod_{k=1}^{K} N_k^{u_k}}\sum_{a_1=1}^{u_1}\cdots\sum_{a_K=1}^{u_K}\sum_{\substack{\forall k,\mathbf{i_k}:\\ \rho(\mathbf{i_k})=a_k}}\mathbb{E}\Big[\prod_{k=1}^{K}\prod_{n=1}^{a_k} Z_n^k\Big]$$

$$= \frac{1}{\prod_{k=1}^{K} N_k^{u_k}}\mathbb{E}\Bigg[\prod_{k=1}^{K}\Big(\sum_{n=1}^{N_k} Z_n^k\Big)^{u_k}\Bigg]$$

$$= \mathbb{E}\Big[\prod_{k=1}^{K}(R_k)^{u_k}\Big]$$

Hence the moments of the joint distribution of $\boldsymbol{Q}$ and $\boldsymbol{R}$ are the same, therefore the joint distributions must agree. □

*Proof of Causal de Finetti.* Recall $\{(X_{1;n}, X_{2;n}, \ldots X_{d;n})\}_{n\in\mathbb{N}}$ is an infinite exchangeable sequence and satisfies condition 2 in Causal de Finetti. Without loss of generality, we reorder the variables according to reversed topological ordering, i.e. a node's parents will always be placed after the node itself. Note a reversed topological ordering is not unique, but it must satisfy a node's non-descendants will come before itself. Then by Kolmogorov's chain rule, we can always write the joint probability distribution as

$$P(\{(X_{1;n}, X_{2;n}, \ldots X_{d;n})\}_{n=1}^{N}) = \prod_{i=1}^{d} P(X_{i;:}|UP_{i;:}) \tag{19}$$

Recall $UP_{i;:} := (X_{i+1;:}, \ldots, X_{d;:})$, which contains all random variables that have higher variable index value than $i$, i.e. upstream of node $i$.

For each $X_{i;:}$, we want to show there exists a suitable probability measure $\nu_i$ such that we can write $P(X_{i;:}|UP_{i;:}) = \int\prod_{n=1}^{N} p(X_{i;n}|PA_{i;n}, \boldsymbol{\theta_i})d\nu_i(\boldsymbol{\theta_i})$. This has been shown in Theorem 7. Hence the

joint distribution becomes:

$$P(\{(X_{1;n}, X_{2;n}, \dots X_{d;n})\}_{n=1}^{N})$$

$$= \int \prod_{n=1}^{N} \prod_{i=1}^{d} p(X_{i;n}|PA_{i;n}, \boldsymbol{\theta_i}) d\nu_i(\boldsymbol{\theta_i})$$

and we complete the proof. $\qquad\qquad\square$

## C   Proof of Identifiability result

### C.1   Identifiability under i.i.d

Let's first see why under i. i. d regime, it is only possible to differentiate the causal structure up to a Markov equivalence class.

**Definition 11** ($\mathcal{I}$-map)**.** *Let $P$ be a distribution, $\mathcal{I}(P)$ denotes the set of conditional independence relationships of the form $X \perp\!\!\!\perp Y \mid Z$ that hold in $P$. Let $\mathcal{G}$ be a DAG, $\mathcal{I}(\mathcal{G})$ denotes the set of conditional independence assumptions encoded in $\mathcal{G}$ which can be directly read-off via d-separation (Pearl, 1988).*

**Definition 12** (Bayesian network structure)**.** *A Bayesian network structure $\mathcal{G}$ is a directed acyclic graph whose nodes represent random variables $X_1, \dots, X_n$. Let $PA_i^{\mathcal{G}}$ denotes the parents of $X_i$ in $\mathcal{G}$, and $ND_i^{\mathcal{G}}$ denotes the variables in the graph that are not descendants of $X_i$.*

**Definition 13** (Global markov property)**.** *Given a DAG $\mathcal{G}$ and a joint distribution $P$, this distribution is said to satisfy **global markov property** with respect to the DAG $\mathcal{G}$ if $\mathcal{I}(\mathcal{G}) \subseteq \mathcal{I}(P)$ (Pearl, 2009). Alternatively, we say $P$ is **Markovian** with respect to $\mathcal{G}$.*

**Definition 14** (Faithfulness)**.** *Given a DAG $\mathcal{G}$ and a joint distribution $P$, $P$ is **faithful** to the DAG $\mathcal{G}$ if $\mathcal{I}(P) \subseteq \mathcal{I}(\mathcal{G})$ (Pearl, 2009).*

We denote $\mathcal{M}(\mathcal{G})$ to be the set of distributions that are Markovian and faithful with respect to $\mathcal{G}$:

$$\mathcal{M}(\mathcal{G}) := \{P : \mathcal{I}(P) = \mathcal{I}(\mathcal{G})\}$$

Two DAGs $\mathcal{G}_1, \mathcal{G}_2$ are Markov equivalent if $\mathcal{M}(\mathcal{G}_1) = \mathcal{M}(\mathcal{G}_2)$.

**Lemma 9** (Graphical criteria for Markov Equivalence (Peters et al., 2017))**.** *Two DAGs $\mathcal{G}_1$ and $\mathcal{G}_2$ are Markov equivalent if and only if they have the same skeleton and the same v-structures.*

This means for any suitably $i.i.d$ generated distribution $P$, one cannot uniquely determine the underlying graph that generates this distribution but can only determine up to d-separtion equivalence.

### C.2   Identifiability under exchangeable

**Definition 15** (Acyclic Directed Mixed Graph (ADMG))**.** *A graph $\mathcal{M}$ is acyclic if it contains no directed cycles, i.e a sequence of edges of the form $x \rightarrow \cdots \rightarrow x$. There are two types of edge between a pair of vertices in ADMG: directed ($x \rightarrow y$) or bi-directed ($x \leftrightarrow y$). In particular, there could be two edges between a pair of vertices, but in this case at least one edge must be bi-directed to avoid a directed cycle.*

**Definition 16** (ICM operator on a DAG)**.** *Let $U$ be the space of all DAGs whose nodes represent $X_1, \dots, X_d$. Let $V$ be the space of ADMGs whose nodes represent $\{(X_{i;n})\}$, where $i \in [d], n \in \mathbb{N}$. A mapping $F$ from $U$ to $V$ is an ICM operator if $F(\mathcal{G})$ satisfies:*

- *$F(\mathcal{G})$ restricted to the subset of vertices $\{X_{1;n}, \dots, X_{d;n}\}$ is a DAG $\mathcal{G}$, for any $n \in \mathbb{N}$,*

- *$X_{i;n} \leftrightarrow X_{i;m}$ whenever $n \neq m$ for all $i \in [d]$*

- *there are no other edges other than stated above*

*We denote the resulting ADMG as $ICM(\mathcal{G})$. Let $\boldsymbol{PA}_{i;n}^{\mathcal{G}}$ denote the parents of $X_{i;n}$ in $ICM(\mathcal{G})$ and similarly for $\boldsymbol{ND}_{i;n}^{\mathcal{G}}$ for corresponding non-descendants.*

**Theorem 9** (Markov equivalence criterion for ADMGs (Spirtes and Richardson, 1997)). *Two ADMGs over the same set of vertices are Markov equivalent if and only if*

1. *They have the same skeleton*

2. *They have the same v-structures*

3. *If a path u is a discriminating path for a vertex B in both graphs, then B is a collider on the path in one graph if and only if it is a collider on the path in the other.*

*Here we do not specify the details for condition 3 as it is not used in below proofs, for more details please refer to (Spirtes and Richardson, 1997).*

We inherit the argument used in finding correct causal structure in the i.i.d. regime where we match conditional independence assumptions encoded in the graph with observed conditional independence relationships in distributions.

**Corollary 2.**

$$\mathcal{I}(ICM(\mathcal{G}_1)) = \mathcal{I}(ICM(\mathcal{G}_2)) \Leftrightarrow \mathcal{G}_1 = \mathcal{G}_2$$

*Proof.* The direction for the case $\mathcal{G}_1 = \mathcal{G}_2$ is trivial. We are left to show that when $\mathcal{G}_1 \neq \mathcal{G}_2$, $\mathcal{I}(ICM(\mathcal{G}_1)) \neq \mathcal{I}(ICM(\mathcal{G}_2))$.

Suppose $\mathcal{G}_1, \mathcal{G}_2$ are BNs over the same set of random variables with length $n$. Suppose further $\mathcal{G}_1 \neq \mathcal{G}_2$, and is not markov equivalent, i.e. $\mathcal{I}(\mathcal{G}_1) \neq \mathcal{I}(\mathcal{G}_2)$. Be definition of $ICM(\mathcal{G})$, we have $\mathcal{I}(\mathcal{G}) \subseteq \mathcal{I}(ICM(\mathcal{G}))$ .Thus $\mathcal{I}(\mathcal{G}_1) \neq \mathcal{I}(\mathcal{G}_2)$ implies $\mathcal{I}(ICM(\mathcal{G}_1)) \neq \mathcal{I}(ICM(\mathcal{G}_2))$.

Consider the case $\mathcal{G}_1 \neq \mathcal{G}_2$, but they are Markov equivalent. We know two DAGs are Markov equivalent if and only if they have the same skeleton and same v-structures. Thus, $\mathcal{G}_1, \mathcal{G}_2$ differs in some node $X_i$ where the orientation of its edge to some node $X_j$ is different while not deleting or creating new v-structures within $\mathcal{G}_1$ and $\mathcal{G}_2$. Wlog, let $X_{i;n} \to X_{j;n}$ in $\mathcal{G}_1$ and $X_{i;n} \leftarrow X_{j;n}$ in $\mathcal{G}_2$. Note $ICM(\mathcal{G}_1)$ and $ICM(\mathcal{G}_2)$ have different v-structures: $X_{i;n} \to X_{j;n} \leftrightarrow X_{j;n+1}$ is a v-structure in $ICM(\mathcal{G}_1)$, but not in $ICM(\mathcal{G}_2)$ as $X_{i;n} \leftarrow X_{j;n} \leftrightarrow X_{j;n+1}$. Thus by Theorem 9, result follows. □

Denote $\mathcal{E}(\mathcal{G})$ to be the set of distributions that are Markovian and faithful to $ICM(\mathcal{G})$:

$$\mathcal{E}(\mathcal{G}) := \{P : \mathcal{I}(P) = \mathcal{I}(ICM(\mathcal{G}))\}$$

Two DAGs $\mathcal{G}_1, \mathcal{G}_2$ are Markov equivalent under ICM generative process if $\mathcal{E}(\mathcal{G}_1) = \mathcal{E}(\mathcal{G}_2)$. By Corollary 2, $\mathcal{E}(\mathcal{G}_1) = \mathcal{E}(\mathcal{G}_2)$ if and only if $\mathcal{G}_1 = \mathcal{G}_2$.

# D  Algorithm

## D.1  Proof

Assume there exists no unobserved latent variables and our observed data is indeed generated from some ICM generative process. Algorithm 1 can identify the underlying DAG. Below we show its main steps.

1. Identify the topological ordering of observed variables

2. Identify edges between different topological orders.

**Definition 17** ($n$-order sinks). *Given a DAG, $X_{i;n}$ denotes the $n$-th sample in one environment and $i$-th variable in $n$-th sample:*

- *A node $X_{i;n}$ is a first-order sink $S_1$ if it does not have any outgoing edges.*

- *A node $X_{i;n}$ is an $k + 1$-order sink $S_{k+1}$, if all of its outgoing edges are to $l$-order sink (where $l < k + 1$) and at least one of them is a $k$-order sink.*

*We denote the set of $k$-order sinks as $S_k$ and $\cup_{i=1}^{k-1} S_i = S_{<k}$.*

---

**Algorithm 1** "Causal-de-Finetti" Algorithm: causal discovery in ICM-generative processes

---

**Input:** For $e \in \mathcal{E}$, we have $(X_{1;n}^e, \ldots, X_{d;n}^e)_{n=1}^{N_e}$ where $X_{i;n}^e$ denotes the $i$-th variable of $n$-th sample in environment $e$ and $N_e$ is the number of samples in environment $e$. Assume $N_e \geq 2, \forall e$.

**Output:** A directed acyclic graph $\mathcal{G}$

**Step 1:** Identify variables' topological ordering. Initiate index list $L := [1, \ldots, d]$ and an empty dictionary of lists $\mathbf{S}$ with keys from $L$. Set starting index $k = 1$.

**while** $L$ *is not empty* **do**
    **for** $i \in L$ **do**
        **if** $X_{i;1}^e \perp\!\!\!\perp X_{j;2}^e \mid \{X_{k;1}^e\}_{k \neq i}, \forall j \neq i$ **then**
            Append $i$ in $\mathbf{S}[k]$ and remove $i$ from $L$
        **end**
    **end**
    $k = k + 1$
**end**

**Step 2:** Identify edges. Set $t := 1$ and reset $k := 1$. Here since each sample shares the same underlying causal graph, we abbreviate $X_{i;n}$ with $X_i$ where $n$ can be any number.

**while** $t \leq d - 1$ **do**
    **while** $k \leq d - t - 1$ **do**
        **for** *each* $i \in S_k$ *and* $j \in S_{k+t}$ **do**
            **if** $t = 1$ **then**
                Test $X_i \perp\!\!\!\perp X_j \mid \bigcup_{m>k+t} \bigcup_{i \in S_m} X_i$, if it holds then there exist no edge, else $X_{j;n} \to X_{i;n}, \forall n$.
            **end**
            **if** $t > 1$ **then**
                Define Z contains three set of nodes: $\bigcup_{m>k+t} \bigcup_{i \in S_m} X_i$, $\mathbf{PA}_i \cap S_{<k+t}$, $S_{k+t} \backslash X_j$.
                Test $X_i \perp\!\!\!\perp X_j \mid Z$, if it holds then there exists no edge, else $X_{j;n} \to X_{i;n}, \forall n$.
            **end**
        **end**
    **end**
**end**

---

In Step 1 of the algorithm, we aim to use appropriate conditional independence tests to determine the topological ordering of our observed variables. The main idea is we iteratively find the first-order sinks $S_1$ and then remove them to find the next first-order sinks, and so on.

**Lemma 1.** *A node $X_{i;n} \in S_1$ if and only if for every $m \neq n$ and $j \neq i$, $X_{i;n} \perp\!\!\!\perp X_{j;m} \mid \{X_{k;n}\}_{k \neq i}$.*

*Proof of Lemma 1.* Let $X_{i;n} \in S_1$. We first examine all the possible paths between $X_{i;n}$ and $X_{j;m}$. There are two cases for any such paths: it starts by $X_{i;n} - X_{k;n} - \ldots$ for some $k \neq i$, or $X_{i;n} - \theta_i - X_{i;p}$ for some $p \neq n$. In the first case, since $X_{i;n}$ is a first-order sink, the first edge is outgoing from $X_{k;n}$ and hence is blocked by conditioning on $X_{k;n}$. In the second case, we cannot continue from $X_{i;p}$ since it does not have any outgoing edges and the path would then include a collider and we thus did not condition on $X_{i;p}$.

To prove the converse, assume that $X_{i;n}$ is not a first-order sink but still satisfies the conditional independence. However, this would mean that $X_{i;n}$ has an outgoing edge to $X_{k;n}$ for some $k \neq i$. Then the path $X_{i;n} \to X_{k;n} \leftarrow \theta_k \to X_{k;m}$ is activated by conditioning on $X_{k;n}$, and hence the conditional independencies cannot hold. $\qquad\square$

This lemma provides us with a test to search for first-order sinks. We can state a similar lemma for $k$-order sinks (note that conceptually the lemma is equivalent to iteratively find first order sinks after removing the original lower order sinks).

**Lemma 10** ($k$-order sink condition). *A node $X_{i;n}$ is a $k$-order sink if and only if the following holds for every $m \neq n$ and $j \neq i$ and $X_{j;m} \in S_{\geq k}$:*

$$X_{i;n} \perp\!\!\!\perp X_{j;m} \mid \{X_{l;n}\}_{l \neq i} - S_{<k}$$

Using Lemma 1 and 10, we can iteratively determine the set of $S_k$ for all $k$. Note, that these sets represent a topographic ordering of observable variables: edges can only run from higher order sinks

to lower order sinks. Hence, the only thing remaining is to determine for every $k$ and every pair of nodes $X_{i;n} \in S_k$ and $X_{j;n} \in S_{<k}$ if they are connected by an edge. Since each sample shares the same underlying causal structure, without loss of generality, we abbreviate $X_{i;n}$ with $X_i$.

**Lemma 2.** *Let node $X_i \in S_n$ and $X_j \in S_m$ where $m < n$. Set $k := n - m$. There does not exist a directed edge from $X_i$ to $X_j$ if and only if when $k = 1$, $X_i \perp\!\!\!\perp X_j \mid S_{>n}$; and when $k > 1$: $X_i \perp\!\!\!\perp X_j \mid Z$, where $Z = S_{>n} \cup (\mathbf{PA}_j \cap S_{<n}) \cup (S_n \backslash X_i)$.*

*Proof of Lemma 2.* First, we prove direction ($\Rightarrow$) and suppose there is no direct edge, i.e. $X_i \to X_j$. If there is no path connecting $X_i$ and $X_j$, then $X_i \perp\!\!\!\perp X_j$. The result trivially holds. Next, we assume there is a path $p$ connecting $X_i$ and $X_j$. The path must satisfy one of the two conditions: Either (case 1) there exists $X_k$ in the path $p$ such that $X_k \in S_{>n}$, or (case 2) all variables in the path $\in S_{\leq n}$.

When $k = 1$: Under case 1), let $W$ be the set containing all variables belong to $S_{>n}$. Note $|W| \geq 1$ by condition. Then there exists a non-collider $X_k \in W$. Suppose all variables in $W$ are colliders and $W \neq \emptyset$, then there must $\exists X_l \in p$ and $X_l \notin W$ that has an edge outgoing to some variable contained in $W$. Since $W \neq \emptyset$, call that variable as $X_w$ and $X_w \in S_{>n}$. By definition of sink orders, $X_l \in S_{>n+1}$, $X_l \in W$. So $W$ is incomplete. Contradiction. Under case 2), we show that there exists a collider in the path. Suppose there is no collider on the path, then all the edge are directed edges in one direction. Since edges can only go from higher sink orders to lower sink orders, the edges can only go from $X_i$ to $X_j$ in one direction. Since the path is also not a 1-arrow direct path, there $\exists X_a \in p, a \neq \{i, j\}$. Choose $X_a \in \mathbf{PA}_j$, since it has a 1-arrow direct path to $X_j$, $X_a \in S_{\geq m+1}$. Further since $X_i$ is an ancestor of $X_a$, $X_i \in S_{\geq m+2}$. Here $n = m + 1$, so $X_i \in S_{\geq n+1}$. Contradicts the condition that $X_i \in S_n$. Given the path connecting $X_i$ and $X_j$ either contains variables in higher sink orders and there exist a non-collider $X_k \in p$ and $X_k \in S_{>n}$, such path can be blocked by conditioning on the set $S_{>n}$; or the path only contains variables in $S_{\leq n}$ and the path must exist a collider, which we block the path by not conditioning on any variables in $S_{\leq n}$. Hence the proposed conditional independence holds by d-separation. By Markov assumption, the conditional independence also holds in distribution.

when $k > 1$: Similarly, under case 1), by same argument as above, there exist a non-collider $X_k \in p$ and $X_k \in S_{>n}$. Therefore conditioning on all variables in $S_{>n}$ blocks the set of paths under case 1. Under case 2) when all variables in the path belong to $S_{\leq n}$, then the parent of $X_j$ in this path, here we call it $X_p$, either (case 2.1) $X_p \in \mathbf{PA}_j \cap S_{<n}$, or (case 2.2) $X_p \in \mathbf{PA}_j \cap S_n$. Under case 2.1, $X_p$ is a non-collider in the path, since it has one outgoing edge. Then conditioning on $\mathbf{PA}_j \cap S_{<n}$ blocks the set of paths under case 2.1. Under case 2.2), for any variable $X_m \in S_n$ on the path $p$, $X_m$ only has outgoing edges since all variables on the path belong to $S_{\leq n}$. $X_m$ is also a non-collider, thus conditioning on $S_n$ blocks the set of paths satisfying case 2.2). Similarly, the proposed conditional independence holds by d-separation. By Markov assumption, the conditional independence also holds in distribution.

Finally, we prove the other direction ($\Leftarrow$): suppose the proposed conditional independence holds, then there does not exist a 1-arrow direct path from $X_i$ to $X_j$. Suppose there exists a 1-arrow directed path and conditional independence holds, then conditioning on the proposed set does not d-separate the path. By faithfulness, conditional independence does not hold in distribution and the result follows. □

# E   Relation to causality in time-series

Recall temporal SCMs: let $\mathbf{V}_t = (V_t^1, \ldots, V_t^N)$ represent the dynamic process variables underlying a multivariate time series. The structural assignments are:

$$V_t^j := f^j(pa(V_t^j), \eta_t^j), \forall V_t^j \in \mathbf{V}_t \text{ and } t \in \mathbb{Z} \tag{20}$$

with jointly independent random variables $\eta_t^j$. The causal parents $pa(V_t^j)$ are direct causes and a subset of $\{\mathbf{V}_t, \ldots, \mathbf{V}_{t-\tau_{max}}\} \backslash \{V_t^j\}$ with $\tau_{max} \geq 0$.

The non-negative integer $\tau_{max}$ means sequential data has directional influence, i.e., no future events could influence the current. Zero is included, as there may be contemporaneous causal influences $V_t^i \to V_t^j$. One could assume such SCM is causally stationary, i.e., the causal relationships and noise distributions are assumed to be invariant in time.

Consider an ICM-generative process, the sample index is exchangeable, i.e., the order of observation does not matter. This is in contradiction to causality in time series as only earlier or contemporaneous samples could constitute potential causal parents of one variable.

When one interprets the sample index in the ICM-generative process as time steps, for example in bivariate cases. $X_t \rightarrow Y_t, X_{t+1} \rightarrow Y_{t+1}, X_t \leftrightarrow X_{t+1}, Y_t \leftrightarrow Y_{t+1}, \forall t$. Firstly, it follows a degree of causally stationary assumption, i.e., the causal relationships are assumed to be invariant in time. Secondly, all sequences of time series have unobserved confounders that influence all the variables in time series. Sometimes, one may interpret such temporal SCMs as the out-of-variable problem, i.e. one lacks the observation of the unobserved confounder $\theta_t$, if it exists in the physical world. Suppose we observe $\theta_t, \psi_t, \forall t$, then ICM-generative process with sample index as time-steps can be rewritten as $\theta_t \rightarrow X_t \rightarrow Y_t \leftarrow \psi_t, \theta_{t+1} \rightarrow X_{t+1} \rightarrow Y_{t+1} \leftarrow \psi_{t+1}, \theta_t \rightarrow X_{t+1}, \psi_t \rightarrow Y_{t+1}$, with $\theta_t = \theta_0, \psi_t = \psi_0, \forall t$. Thus, it follows the temporal SCM formulation. It suggests data with less structure can be modelled by formulations for data with more complex structure with the right instantiation - this is the case for exchangeable sequences with i.i.d. data, and so is the case for time-series data with exchangeable data.