# OpenReview forum: "Causal de Finetti: On the Identification of Invariant Causal Structure in Exchangeable Data"
_NeurIPS.cc/2023/Conference — NeurIPS 2023 poster_

### Official Review · Reviewer_3rag · 2023-07-05

**Soundness:** 3 good
**Presentation:** 3 good
**Contribution:** 3 good
**Rating:** 7
**Confidence:** 2

**Summary:**

This paper is about causal discovery with exchangeable data, thus relaxing the traditional iid assumption. This allows for causal structure identification by conditional independence tests. The result has application to multi-environment data.

**Strengths:**

Despite my low confidence, I think the paper delivers strong results for causal analysis. Both the Causal De Finetti theorems and the identifiability result (Theorem 5) appear not trivial and of some impact on people working in the field.

**Weaknesses:**

- The identifiability result seems to hold for Markovian models only.
- The experiments are very promising, but only a very limited setup.
- The paper is very technical and dense, thus more suitable for a journal version. Yet, it can be worth presenting such a condensed version at the Neurips conference.



**Questions:**

I have found it surprising that a relaxation of the iid assumption leads to higher identifiability. Is it possible to better explain this point?

Figure 1a could be clearer to me. Is it possible to make it more expressive (or drop it)?

Is there any room to extend the result to non-Markovian models?


**Limitations:**

No specific issues related to that.

---

> ### Author Rebuttal · Authors · 2023-08-09
>
> We thank the reviewer for the efforts and we greatly appreciate your positive feedback. From your comments, you clearly understood the paper.
>
> > The identifiability result seems to hold for Markovian models only.
>
> We thank the reviewer for the suggestion, however, Pearl points out in his classical textbook [1] that “Markov compatibility” is "a necessary and sufficient condition for a DAG G to explain a body of empirical data represented by P, that is, to describe a stochastic process capable of generating P” (Definition 1.2.2). Also the classical result on ‘two DAGs are observationally equivalent’, i.e., one can only distinguish causal structures up to Markov equivalence classes in i.i.d. data, is also established on the condition of Markov compatibility. Here we take the same view and show one can distinguish unique causal structure in exchangeable data. Just as the development of causal discovery in i.i.d. data, we hope there is follow-up work for more practical considerations of non-Markovian models in exchangeable data.
>
> > The experiments are very promising, but only a very limited setup.
>
> We thank the reviewer for appreciating the values of experiments. Indeed, we aim for the experiments to serve as a demonstration for our identifiability theorem, i.e., ICM-generative processes can indeed recover unique causal structure via conditional independence tests only, and current methods developed perform poorly in our setting. We also agree with the reviewer that it would be interesting to extend the experiments further. Here we include experiment results that are compared to more baselines, (specifically, FCI, GES and NOTEARS). We observe that existing methods all perform poorly in our setting. We will include the result and a more detailed evaluation in an updated version.
>
> > The paper is very technical and dense, thus more suitable for a journal version. Yet, it can be worth presenting such a condensed version at the Neurips conference.
>
> We agree with the reviewer that the paper is ‘technical and dense’. We also agree with the reviewer that it is worth presenting in a venue like NeurIPS to demonstrate the advantages of exchangeable data for causality.
>
> > I have found it surprising that a relaxation of the iid assumption leads to higher identifiability. Is it possible to better explain this point?
>
> We thank the reviewer for the question, it is indeed one of the main point of this paper. The higher identifiability power is due to there exist richer conditional independence structures contained in exchangeable data. For example, consider pairs of $(X_1, Y_1)$, $(X_2, Y_2)$. The conditional independence $Y_1 \perp X_2 | X_1$ trivially holds in i.i.d. data, but holds non-trivially in exchangeable data. This thus allows us to identify unique causal structures.
>
> > Figure 1a could be clearer to me. Is it possible to make it more expressive (or drop it)?
>
> We thank the reviewer for the suggestion and will try to incorporate it in a future version.
>
> > Is there any room to extend the result to non-Markovian models?
>
> We thank the reviewer for the question and will illustrate the potential of extension in a simple example of the bivariate case where hidden confounders are allowed. Suppose there are two pairs of variables $(X_1, Y_1, Z_1)$, $(X_2, Y_2, Z_2)$, where $Z_1, Z_2$ are unobserved confounders. It follows the causal structure, $X_i \to Y_i, Z_i \to X_i, Z_i \to Y_i$, for all $i = \{1, 2\}$. In particular, $(X_i, Y_i)$ are generated by an exchangeable process connected by latent variables $\theta$, $\psi$ and $Z_i$ is generated by an i.i.d. process. Note the conditional independence $Y_2 \perp X_1 | X_2$ still holds in this setting where there exists hidden confounders. This suggests one can still recover the causal direction of $X$ and $Y$ in this case.
>
> Reference:
> 1. Pearl, J. (2009). Causality. Cambridge University Press.

---

> > ### Comment · Reviewer_3rag · 2023-08-16
> > **Keeping the same (positive) recommendation**
> >
> > I thank the authors for their comments, which helped me understand some points better. As far as I can understand, the (serious) problem with the proof of Lemma 2 raised by Reviewer ikix has been fixed, and this had no impact on the empirical validation of the result. In this situation, I am happy to confirm my positive (but low confidence) opinion about the paper.

---

> > > ### Author Response · Authors · 2023-08-21
> > >
> > > We thank the reviewer for responding and their positive feedback on the paper.

---

### Official Review · Reviewer_ikix · 2023-07-06

**Soundness:** 3 good
**Presentation:** 2 fair
**Contribution:** 3 good
**Rating:** 6
**Confidence:** 4

**Summary:**

Causal analogs to de Finetti's theorem are proven, showing that if in an exchangeable distribution certain conditional independences hold, the distribution can be seen as being generated by a DAG with a latent variable corresponding to each node, determining that node's causal mechanism. It is further shown that knowing these conditional independences allows unique identification of the DAG. A causal discovery algorithm is presented leveraging these results, and evaluated in a synthetic data setting where data come from very many different environments.

**Strengths:**

* The causal de Finetti's theorems are philosophically interesting, similar to how the original de Finetti's theorem can play a role in the justification of Bayesian inference. The resulting graphical models, expressing disentangled causal mechanisms in terms of latent variables (Figure 1b, right), are very insightful and deserve to be commonly known in the community.

* The resulting conditional independences, which are testable if data are available from multiple environments, are a very useful ingredient for causal discovery algorithms in such settings.

**Weaknesses:**

* The related work section only considers causal discovery, not the causal de Finetti theorem itself. Such references should also be listed, because the paper is claiming a contribution in this area. For instance, [Dawid 2021] also uses exchangeability to do causal inference.

  * Also about the related work section: the sentence "These algorithms on non-i.i.d. grouped data all demonstrate success, though it is unclear why grouped data enable causal structure identification." - I disagree strongly with this, there is a very good understanding of why multiple environments (possibly including interventional data) help with causal discovery, in the papers you list as well as in the causal inference literature as a whole.

* Implications of the causal de Finetti theorem are listed without sufficiently arguments, and I believe overstated. See question about line 139 below.

* Here is a counterexample to Lemma 2: $X_i \rightarrow X_a \rightarrow X_b \leftarrow X_c \leftarrow X_d \rightarrow X_j$ with $X_b  \rightarrow X_j$. Then $X_d \in S_n$, so only $X_b$ is conditioned on, but this opens the listed path.
(Separate from this, in line 294 explaining the lemma, I think "non-directed" should be "open": also directed paths other than the 1-arrow path should be blocked.)

* Experiment (some things are unclear to me now, see questions below):

  * other methods are not really fit for this scenario

  * 1 x-y pair per environment (?): this disconnects the experiment from the theory

**References:**

[Dawid 2004]: Probability, Causality and the Empirical World: A Bayes-de Finetti-Popper-Borel Synthesis, Statistical Science , Feb., 2004, Vol. 19, No. 1 (Feb., 2004), pp. 44-57

[Dawid 2021]: Decision-theoretic foundations for statistical causality, Journal of Causal Inference 2021; 9: 39-77

**Questions:**

* Line 139: "one can separately manipulate each latent variable controlling different mechanisms" - Does this follow from equation 5? How? ([Dawid 2004] warns that de Finetti's theorem only establishes that the latent variable exists in our minds, not in the real world.)

* In the experiment, I assume $\tilde{N}^e$ is a 2-element vector? What about $N^e$? And how many x-y pairs are sampled per environment?

Suggestions to improve the language (not relevant for my assessment of the paper):

* the spelling of "i.i.d." is inconsistent (sometimes with spaces, sometimes with the final . missing)

* line 92 & 595: "infinite exchangeable" -> "infinitely exchangeable"

* the final sentence of section 2.2 doesn't parse ("due to"&"that underlies"; "involving observations are i.i.d.")

* below definition 2: "does not hold for all" -> "does not hold for any"

* line 140 & 141: "supporting mechanisms" -> "supporting that mechanisms" (2x)

* line 147: "one implicitly make" -> "makes". Similar in appendix A.

* Theorem 3 & 4: I'd replace "The sequence is" by "If", add "the sequence is" to point 1, and remove "if" from point 2

* line 201: "decide" -> "deciding"

* line 220: "process" -> "processes" (also elsewhere)

* Definition 5: "Given $P$ is" -> "Let $P$ be"; "Given $G$ be" -> "Let $G$ be"; "a ADMG" -> "an ADMG" (this also appears in Def 6); "read-off" -> "read off"

* Definition 7: There is only one mapping fitting this definition, so define it straightaway instead of defining when something "is an ICM operator". Line 242 has a double "other"

* line 280: $<$ -> $\leq$

* line 287: "topological ordering" is not the right concept here, as that is a total order

* line 629: "exchangeable" -> "exchangeability"

* line 679: "$l < k + 1$ -> $l \leq k$

**Limitations:**

Yes (except as discussed above)

---

> ### Author Rebuttal · Authors · 2023-08-09
>
> We thank the reviewer for the thorough feedback, and we appreciate the kind words that causal de Finetti theorems “deserve to be commonly known in the community”. Thank you for pointing out the interesting reference Dawid 2021. We will discuss and cite [1] in an updated version.
>
> Re ‘grouped data’: This paper meant to express that there is work [2] assuming ICM, but does not come with an understanding for its implications on data’s probabilistic relationships. We thank you for your feedback and we promise to rephrase the sentence and clarify that our contribution is with respect to the statistical understanding of ICM.
>
> Re de Finetti parameters: We are sorry we failed to clarify this. We did not mean to claim that the variable necessarily exists in the real world - only that the conditions of Thm. 2 imply that the data looks as if it has been generated by a process with independent mechanisms. (We are no experts on those philosophical ramifications, so we try to focus on technical results.)
>
> Re counterexample to Lemma 2 (only used for structure learning, not affecting causal de Finetti theorems and identifiability theorem): We thank the reviewer for spotting this error. Here we correct the lemma and its proof. For its impact on the empirical evaluations, we re-run the experiments with the updated fix and observe no changes in the results (see the uploaded pdf). We will include all the changes in the updated version.
>
> **Lemma 2** Let node $X_i \in S_n$ and $X_j \in S_m$ where $m < n$. Set $k:= n-m$. There does not exist a directed edge from $X_i$ to $X_j$ iff when $k = 1$,  $X_i \perp X_j | S_{>n}$; and when $k > 1$: $X_i \perp X_j | Z$, where $Z = S_{>n} \cup (PA_j \cap S_{<n}) \cup {S_n \backslash X_i}$.
>
> **Proof of Lemma 2**: ($\Rightarrow$): suppose there is no direct edge, i.e. $X_i \to X_j$. If there is no path connecting $X_i$ and $X_j$, then $X_i \perp X_j$. Next, we assume there exists a path $p$ connecting $X_i, X_j$. Then $p$ satisfies: Either (1) there exists $X_k \in p$ s.t. $X_k \in S_{>n}$, or (2) all variables in the path $\in S_{\leq n}$.
>
> When $k = 1$:
>     Under (1), let $W$ be the set containing all variables $\in S_{>n}$. Note $|W| \geq 1$. Then there exists a non-collider $X_k \in W$. Suppose all variables in $W$ are colliders, then there must $ \exists X_l \in p$ and $X_l \not \in W$ that has an edge outgoing to some variable contained in $W$. By definition of sink orders, $X_l \in S_{>n+1} \implies X_l \in W$. Contradiction.
>     Under (2), we show that there exists a collider in the path. Suppose there is no collider on the path, then all the edges point in one direction, here the edges flow from $X_i$ to $X_j$. Since the path is not a 1-arrow direct path, there $\exists X_a \in p, a \neq \{i, j\}$.
> Let $X_a$ be the neighbour of $X_j$. If $X_a \in PA_j$, then $X_a \in S_{\geq m + 1}$. As $X_i$ is an ancestor of $X_a$, $X_i \in S_{\geq m+2} = S_{\geq n+1}$. If $X_a \in CH_j$, if edges flow in one direction, $X_j$ is an ancestor of $X_i$. Contradiction.
>     Given the path connecting $X_i$ and $X_j$ either contains variables in higher sink orders and there exists a non-collider $X_k \in p$ and $X_k \in S_{>n}$., such path can be blocked by conditioning on the set $S_{>n}$; or the path only contains variables in $S_{\leq n}$ and the path must exist a collider, which we block the path by not conditioning on any variables in $S_{\leq n}$.
>
> when $k > 1$:
>     Under (1), similar as above, there exists a non-collider $X_k \in p$ and $X_k \in S_{>n}$. Therefore conditioning on all variables in $S_{>n}$ blocks the set of paths under (1). Under (2) when all variables in the path belong to $S_{\leq n}$, then let $X_p$ be the parent of $X_j$ in this path, either (2.1) $X_p \in PA_j \cap S_{<n}$, or (2.2) $X_p \in PA_j \cap S_n$. Under (2.1), $X_p$ is a non-collider in the path, since it has one outgoing edge. Then conditioning on $PA_j \cap S_{<n}$ blocks the set of paths under (2.1). Under (2.2), for any variable $X_m \in S_n$ on the path $p$, $X_m$ only has outgoing edges since all variables on the path belong to $S_{\leq n}$. $X_m$ is also a non-collider, thus conditioning on $S_n$ blocks the set of paths satisfying (2.2).
> By Markov assumption, CIs holds in distribution.
>
> ($\Leftarrow$): suppose the proposed CI holds and $\exists X_i \to X_j$, then $X_i$ is not d-separated by $X_j$. By faithfulness, CI does not hold in distribution.
> **End of Proof**
>
> Re clarifications on experiments, here the equations listed represent the data-generating process for one data instance $X^e$. Specifically, $N^e$ is a vector with size $2$ and $\tilde{N}^e$ is also a vector with size $2$. In practice, we generate two instances per environment as stated in line 319, where we fix $N^e$ per environment, and generate $2$ $\tilde{N}^e$ based on the $N^e$ generated and consequently generate 2 data instances $X^e$ based on corresponding $\tilde{N}^e$. We acknowledge that most other methods are designed to tackle data sampling from i.i.d. setting. To this end, we include a comparison with the method ‘CD-NOD’ [2] which is designed for heterogeneous and nonstationary datasets. We also observe CD-NOD performs poorly in our setting.
>
> Overall, we thank you for the detailed review, which we will carefully take into account your feedback on the clarity of the presentation in experimental design, the related work discussion and implications of causal de Finetti theorem, and will update with more detailed clarification as discussed above. We hope that our response has addressed all your questions, especially on the soundness of our paper (with the corrected Lemma). We kindly ask you to let us know if you have any remaining criticism, and - if we have answered your questions - to consider reevaluating your score.
>
> References:
> 1. Dawid, P. (2021). Decision-theoretic foundations for statistical causality.
> 2. Huang, B. et al. (2020). Causal discovery from heterogeneous/nonstationary data.

---

> > ### Comment · Reviewer_ikix · 2023-08-17
> >
> > I think this paper has the potential for high impact, but with this comes great responsibility to be precise and complete about positioning w.r.t. related work. I am happy that the authors promise to improve the paper in this regard. In the case of Lemma 2, they provide a concrete fix, and I agree that the new version of the lemma is correct; I am glad this had no impact on the experimental results.
> >
> > I am raising my score to 6. Not having seen the updated version, I am hesitant about raising it further.

---

> > > ### Author Response · Authors · 2023-08-17
> > >
> > > We thank the reviewer for going the extra mile in finding an error and checking the corrected version. We thank the reviewer also for acknowledging the changes we made and for raising the score. We will take particular care regarding the related work section for the final version, and hope the updated version will not disappoint. In case you feel comfortable revealing your identity after the end of the reviewing process, we would be happy to add your name to the acknowledgement.

---

### Official Review · Reviewer_idLT · 2023-07-06

**Soundness:** 4 excellent
**Presentation:** 3 good
**Contribution:** 3 good
**Rating:** 7
**Confidence:** 4

**Summary:**

The authors describe how the combination of assumptions about exchangeability and specific statistical tests can enhance the causal implications that can be derived from observational data.

**Strengths:**

The utility and implications of exchangeability for causal inference are well described by the authors. Under the assumptions that they outline (the availability of multiple environments), the additional causal implications that can be derived are clearly very useful.

The authors help readers by following essentially all complex mathematical statements with an informal statement of the result in relatively plain language.

**Weaknesses:**

A 2019 paper by Jensen et al. [1] on exchangeability and structure learning in causal graphical models makes a very similar point as this paper regarding the value of multi-environment data and exchangeability. That paper describes how to use knowledge of environments (what that paper calls “parent objects”) to provide additional constraints on graph structure by conditioning on environment (and the latent variables implied by that environment). The proofs in that paper rely on exchangeability and contrast exchangeability with conditional independence. That paper differs somewhat from this paper in goals and focus, but the current paper would be improved by clearly describing the contributions of that paper and delineating the unique contributions of the present paper.

The authors cite the Independent Causal Mechanism principle without reference to the long history of the concept, stretching back almost a century. Pearl’s work [2] (in which it is sometimes called “modularity”) cites an earlier review by Aldrich [3] (in which it is called “autonomy”), which cites very early work by Haavelmo and others as far back as the 1930s. This history is worth at least noting.

The difference between “environments” and random variables is not clearly explained until Section 4, and this distinction is critical to understanding how (for example) schools differ from students in the second example of Section 3.  Each school corresponds to many students. The plate notation introduced in Section 4 would help readers if introduced earlier.

*References*

1. Jensen, D., Burroni, J., & Rattigan, M. (2020, August). Object conditioning for causal inference. In *Uncertainty in Artificial Intelligence* (pp. 1072-1082).

2. Pearl, J. (2009). *Causality*. Cambridge University Press.

3. Aldrich, J. (1989). Autonomy. *Oxford Economic Papers*, *41*(1), 15-34.

**Questions:**

(none)

**Limitations:**

The authors do not sufficiently emphasize the special structure of the data (specifically, multiple environments) that is needed to apply the methods that they advocate. Judging from the results shown in Figure 3, the benefit to multivariate structure learning only becomes substantial at truly large numbers of environments. The need for multiple environments (and large numbers of them) should be emphasized more clearly earlier in the paper.

---

> ### Author Rebuttal · Authors · 2023-08-09
>
> We thank the reviewer for taking the time and are glad to find the reviewer appreciates the “utility and implications of exchangeability for causal inference” and considers our results as “clearly very useful”.
>
> > That paper differs somewhat from this paper in goals and focus, but the current paper would be improved by clearly describing the contributions of that paper and delineating the unique contributions of the present paper.
>
> Thanks for pointing out the work from Jensen et al. [1] and their interesting findings: the probabilistic implications for conditioning on objects can be explained using exchangeability. They show object conditioning, due to exchangeability, has the advantage of mitigating latent confounding and measurement errors for causal inference.  Our work differs from [1] as rather than inference, we show the advantage of exchangeable data for causal discovery and advocate the widely-used causality principle ICM can be expressed as Equation 5.
> Our unique contribution is that researchers often assume ICM principle (or its variants) on the data-generating process, but do not come with an understanding for the implications of assuming ICM, i.e., the probabilistic relationships of the underlying data. Causal de Finetti, explicitly states, the implicit probabilistic assumptions are exchangeability of data and certain conditional independences. In summary, this paper’s unique contribution is to 1) show exchangeable data is better than i.i.d. data in unique causal structure identification (which is previously deemed impossible via CI tests [2] ) and 2) prove causal de Finetti theorems which explicitly connect causality and probabilistic modelling. We thank the reviewer for the suggestion in comparing related work on exchangeability for casual inference and will include more detailed discussions in an updated version.
>
> > The authors cite the Independent Causal Mechanism principle without reference to the long history of the concept, stretching back almost a century. Pearl’s work [2] (in which it is sometimes called “modularity”) cites an earlier review by Aldrich [3] (in which it is called “autonomy”), which cites very early work by Haavelmo and others as far back as the 1930s. This history is worth at least noting.
>
> We thank the reviewer for pointing this out and will include the relevant references, for example, Pearl (2009), Aldrich (1989), Hoover (2008), in an updated version.
>
> > The authors do not sufficiently emphasize the special structure of the data (specifically, multiple environments) that is needed to apply the methods that they advocate. The need for multiple environments (and large numbers of them) should be emphasized more clearly earlier in the paper.
>
> We thank the reviewer for the suggestion. The experiments aimed as a proof-of-concept to demonstrate our identifiability theorem in practice. The need for large number of multiple environments arise due to we only use two samples per environment. From the reviewer's feedback, we understand there could be potential misunderstandings. We will incorporate your suggestions (e.g. explaining the difference between ‘environments’ and random variables, moving the plate notation earlier, emphasize the need of large number of environments) in the updated version to ease readers’ understanding.
>
> References:
> 1. Jensen, D., Burroni, J. &amp; Rattigan, M.. (2020). Object Conditioning for Causal Inference. Proceedings of The 35th Uncertainty in Artificial Intelligence Conference, in Proceedings of Machine Learning Research.
> 2. J. Pearl, Causality: Models, Reasoning, Inference, 2nd ed. New York, NY, USA: Cambridge Univ. Press, 2009.
> 3. J. Aldrich, “Autonomy,” Oxford Econ. Papers, vol. 41, no. 1, pp. 15–34, 1989.
> 4. K. D. Hoover, “Causality in economics and econometrics,” in The New Palgrave Dictionary of Economics, S. N. Durlauf and L. E. Blume, Eds., 2nd ed. Basingstoke, U.K.: Palgrave Macmillan, 2008.

---

> > ### Comment · Reviewer_idLT · 2023-08-20
> >
> > I appreciate the authors' careful reading of my review and their thoughtful response. The changes that you outline will improve the paper.
> >
> > One side note: After reading Hoover (2008), I'm not convinced that it is a particularly good reference regarding autonomy/modularity. Hoover certainly mentions the Lucas Critique (a version of the modularity issue), but most of the entry is on other topics. Aldrich (1989) is much more on-point, as are some specific portions of Pearl (2009) (though the discussion of modularity is oddly dispersed throughout that book).
> >
> > I'll increase my rating.

---

> > > ### Author Response · Authors · 2023-08-21
> > >
> > > We thank the reviewer for acknowledging our changes and raising the score. We thank the reviewer for pointing the appropriate references for ICM and will include the more relevant reference, e.g., Pearl (2009), Aldrich (1989) in the updated version.

---

### Official Review · Reviewer_TMn5 · 2023-07-07

**Soundness:** 4 excellent
**Presentation:** 4 excellent
**Contribution:** 4 excellent
**Rating:** 8
**Confidence:** 4

**Summary:**

This paper examines a stronger notion of exchangeability which provides invariant causal structure, i.e., is capable of serving as the basis for causal reasoning. The main contribution is a theorem which states that for pairs of RV with certain conditional independence properties it is always possible to represent them as a mixture of i.i.d. sequences with identical Markov factorization. This idea is further extended to the multivariate case. The authors then connect this notion of causal exchangeability to causal identification and causal discovery. Experimental results are provided for bivariate orientation and causal discovery from multiple datasets which show favorable performance when taking causal exchangeability into account.

**Strengths:**

I found this paper to be a simple, intuitive, and thorough piece of work that provides an interesting new lens on causal inference. The notion of defining what constitutes exchangeable sequences with invariant causal structures is very interesting, and the authors do a very nice job of motivating the problem, providing thorough and well written theorems, and contextualizing them within the context of application. This is a clear description of causal mechanisms from a purely statistical view and an interesting addition to the literature.

**Weaknesses:**

The largest weakness I can see here is the lack of comprehensive empirical evaluations. The evaluations provided are fairly simplistic, limited in scope, and serve more as a demonstration rather than an evaluation finding relative strength and weakness of the proposed method.

**Questions:**

Did you run any other causal structure learning algorithms such as FCI, GES or even optimization based approaches such as NOTEARS? Given the poor performance of PC I'm wondering if this is a weakness shared across all algorithms.

**Limitations:**

Yes.

---

> ### Author Rebuttal · Authors · 2023-08-09
>
> We thank the reviewer for taking the time and providing a thorough review and constructive feedback. We are glad you found the paper interesting as a simple, intuitive, thorough piece of work.
> > The largest weakness I can see here is the lack of comprehensive empirical evaluations. The evaluations provided are fairly simplistic, limited in scope, and serve more as a demonstration rather than an evaluation finding relative strength and weakness of the proposed method.
> > Did you run any other causal structure learning algorithms such as FCI, GES or even optimization based approaches such as NOTEARS? Given the poor performance of PC I'm wondering if this is a weakness shared across all algorithms.
>
> We agree with the reviewer that the empirical evaluations are rather limited. We mainly use it to illustrate unique causal structure learning is feasible for ICM-generative processes. Here we include additional empirical results that compare with FCI, GES and NOTEARS in the uploaded pdf. Our result shows that poor performance is shared across all algorithms evaluated. This is consistent with the main point of this paper that exchangeable data allows unique causal structure identification using conditional independence tests and existing algorithms (designed under i.i.d. data setting) performs poorly in our settings. We thank the reviewer for the suggestion and will include this result and a more detailed empirical evaluation in the updated version.

---

### Author Rebuttal · Authors · 2023-08-09

We thank all the reviewers for their time and effort in providing valuable feedback. We are glad to see all the reviewers have unanimous agreement that this paper delivers solid contributions to the community and some consider it to be “very insightful and deserve to be commonly known in the community”. We also are glad to see several reviewers appreciate this paper’s simple, intuitive explanation of complex mathematical theorems. We thank again for all reviewer’s recognition of the impact of our work and hope this work provides a new lens on causal inference and can trigger follow-up work in the space of causality and exchangeable data.

We respond to all reviewer’s points in detail but wanted to highlight:

* we followed TMn5’s suggestions for **more comprehensive empirical evaluations**. Here we include comparisons with additional benchmarks, e.g. FCI, GES, NOTEARS, in the uploaded pdf. We observe PC’s poor performance is indeed shared across the additional benchmarks - methods mostly designed for i.i.d. data setting performs poorly on exchangeable data in our setting. We will include the result and a more detailed comparison in the updated version.

---

### Decision · Program_Chairs · 2023-09-21

**Decision:**

Accept (poster)

**Comment:**

The reviewers made some great points during the rebuttal. Reviewer ikix identified a problem with Lemma 2, which the authors provided a patch for; Reviewer TMn5 recommended more extensive experimental evaluations, which the authors conducted; Reviewer idLT pointed to a very relevant paper that the authors missed, alongside pointers on how to better position the current paper, which the authors promised to incorporate; Reviewer 3rag asked the very intuitive question "How come a relaxed assumption, iid -> exchangeable, lead to stronger identifiability results", which the authors tried to answer.

Overall, I believe the rebuttal has been very helpful to critically evaluate the paper, and improve its weaknesses.

I will add a few things based on my reading of the paper:

- I agree with the reviewers that the authors need to be careful not to over-claim the role of causal de Finetti to further our understanding of the problem. As Reviewer ikix points out, the role of multiple environments in causal structure discovery has generally been well understood unlike what the authors claim.

- This was not raised by any of the reviewers, but I believe there might be a connection to time-series causal discovery, which might also help answer Reviewer 3rag's question. Let i be the time index in Xi, Yi. Consider the bivariate graph unfolded in time X1->Y1 + X1<-(unobserved confounder)->X2, X2-> Y2 and so on. This is the setting explored by causal de Finetti. I understand the two settings are philosophically very different and there is no fixed order that we know a priori. But in my opinion, one can view lifting from iid to exchangeable setup puts us somewhere between iid and time series. I suggest exploring this potential connection in the camera ready. I think both literatures can benefit from each other starting with a clear exposition here.

- In terms of bivariate discovery, I think the illustration in Figure 1a might be misleading. Because, as the authors note in the paper, one still needs iid samples of exchangeable pairs, which boils down to sampling from multiple environments. The non-identifiability of causal structure in the bivariate case is only from observational data. Thus, the claim here needs to be much more nuanced than it currently is. Specifically, the authors need iid samples from the graph in Figure 2c to detect (X2 __ || __ Y1 | X1) and learn that X->Y. This illustration ignores this very important context and the authors simply claim "Under i.i.d. data, one can only identify (X __ || __ Y), whereas under exchangeable data, one can identify unique bivariate causal structure." PC also does not seem like the right baseline for this problem of learning from multiple environments. I strongly suggest revising at least this claim, and providing this much needed context early on in the paper (introduction).

Having noted these points, which I hope the authors will take into account in their camera-ready, I will recommend acceptance as the connection between exchangeability and causal discovery is definitely worth noting and I did enjoy reading the view that CI statements across samples are informative of the structure, **if one can test them**. Authors need to clearly convey that being able to test these CI statements requires IID samples from exchangeable pairs, which assume access to a different kind of multi-environment data, and the methodology is not directly comparable to basic observational discovery algorithms such as PC.